# Estimating the Unseen: Improved Estimators for Entropy and other Properties

**Gregory Valiant** [*]
Stanford University
Stanford, CA 94305
valiant@stanford.edu

**Paul Valiant** [†]
Brown University
Providence, RI 02912
pvaliant@gmail.com

## Abstract

Recently, Valiant and Valiant [1, 2] showed that a class of distributional properties, which includes such practically relevant properties as entropy, the number of distinct elements, and distance metrics between pairs of distributions, can be estimated given a *sublinear* sized sample. Specifically, given a sample consisting of independent draws from any distribution over at most $n$ distinct elements, these properties can be estimated accurately using a sample of size $O(n/\log n)$. We propose a novel modification of this approach and show: 1) theoretically, this estimator is optimal (to constant factors, over worst-case instances), and 2) in practice, it performs exceptionally well for a variety of estimation tasks, on a variety of natural distributions, for a wide range of parameters. Perhaps unsurprisingly, the key step in our approach is to first use the sample to characterize the "unseen" portion of the distribution. This goes beyond such tools as the Good-Turing frequency estimation scheme, which estimates the total probability mass of the unobserved portion of the distribution: we seek to estimate the *shape* of the unobserved portion of the distribution. This approach is robust, general, and theoretically principled; we expect that it may be fruitfully used as a component within larger machine learning and data analysis systems.

## 1 Introduction

What can one infer about an unknown distribution based on a random sample? If the distribution in question is relatively "simple" in comparison to the sample size—for example if our sample consists of 1000 independent draws from a distribution supported on 100 domain elements—then the empirical distribution given by the sample will likely be an accurate representation of the true distribution. If, on the other hand, we are given a relatively small sample in relation to the size and complexity of the distribution—for example a sample of size 100 drawn from a distribution supported on 1000 domain elements—then the empirical distribution may be a poor approximation of the true distribution. In this case, can one still extract accurate estimates of various properties of the true distribution?

Many real–world machine learning and data analysis tasks face this challenge; indeed there are many large datasets where the data only represent a tiny fraction of an underlying distribution we hope to understand. This challenge of inferring properties of a distribution given a "too small" sample is encountered in a variety of settings, including text data (typically, no matter how large the corpus, around 30% of the observed vocabulary only occurs once), customer data (many customers or website users are only seen a small number of times), the analysis of neural spike trains [15],

---

[*] http://theory.stanford.edu/~valiant/ A portion of this work was done while at Microsoft Research.
[†] http://cs.brown.edu/people/pvaliant/

and the study of genetic mutations across a population[1]. Additionally, many database management tasks employ sampling techniques to optimize query execution; improved estimators would allow for either smaller sample sizes or increased accuracy, leading to improved efficiency of the database system (see, e.g. [6, 7]).

We introduce a general and robust approach for using a sample to characterize the "unseen" portion of the distribution. Without any *a priori* assumptions about the distribution, one cannot know what the unseen domain elements are. Nevertheless, one can still hope to estimate the "shape" or *histogram* of the unseen portion of the distribution—essentially, we estimate how many unseen domain elements occur in various probability ranges. Given such a reconstruction, one can then use it to estimate any property of the distribution which only depends on the shape/histogram; such properties are termed *symmetric* and include entropy and support size. In light of the long history of work on estimating entropy by the neuroscience, statistics, computer science, and information theory communities, it is compelling that our approach (which is agnostic to the property in question) outperforms these entropy-specific estimators.

Additionally, we extend this intuition to develop estimators for properties of pairs of distributions, the most important of which are the *distance metrics*. We demonstrate that our approach can accurately estimate the total variational distance (also known as *statistical distance* or $\ell_1$ distance) between distributions using small samples. To illustrate the challenge of estimating variational distance (between distributions over discrete domains) given small samples, consider drawing two samples, each consisting of 1000 draws from a uniform distribution over 10,000 distinct elements. Each sample can contain at most 10% of the domain elements, and their intersection will likely contain only 1% of the domain elements; yet from this, one would like to conclude that these two samples must have been drawn from nearly identical distributions.

## 1.1 Previous work: estimating distributions, and estimating properties

There is a long line of work on inferring information about the unseen portion of a distribution, beginning with independent contributions from both R.A. Fisher and Alan Turing during the 1940's. Fisher was presented with data on butterflies collected over a 2 year expedition in Malaysia, and sought to estimate the number of *new* species that would be discovered if a second 2 year expedition were conducted [8]. (His answer was "$\approx 75$.") At nearly the same time, as part of the British WWII effort to understand the statistics of the German enigma ciphers, Turing and I.J. Good were working on the related problem of estimating the total probability mass accounted for by the unseen portion of a distribution [9]. This resulted in the Good-Turing frequency estimation scheme, which continues to be employed, analyzed, and extended by our community (see, e.g. [10, 11]).

More recently, in similar spirit to this work, Orlitsky *et al.* posed the following natural question: given a sample, what distribution maximizes the likelihood of seeing the observed species frequencies, that is, the number of species observed once, twice, etc.? [12, 13] (What Orlitsky *et al.* term the *pattern* of a sample, we call the *fingerprint*, as in Definition 1.) Orlitsky *et al.* show that such likelihood maximizing distributions can be found in some specific settings, though the problem of finding or approximating such distributions for typical patterns/fingerprints may be difficult. Recently, Acharya *et al.* showed that this maximum likelihood approach can be used to yield a near-optimal algorithm for deciding whether two samples originated from *identical* distributions, versus distributions that have large distance [14].

In contrast to this approach of trying to estimate the "shape/histogram" of a distribution, there has been nearly a century of work proposing and analyzing estimators for particular properties of distributions. In Section 3 we describe several standard, and some recent estimators for entropy, though we refer the reader to [15] for a thorough treatment. There is also a large literature on estimating support size (also known as the "species problem", and the related "distinct elements" problem), and we refer the reader to [16] and to [17] for several hundred references.

Over the past 15 years, the theoretical computer science community has spent significant effort developing estimators and establishing worst-case information theoretic lower bounds on the sample size required for various distribution estimation tasks, including entropy and support size (e.g. [18, 19, 20, 21]).

The algorithm we present here is based on the intuition of the estimator described in our theoretical work [1]. That estimator is not practically viable, and additionally, requires as input an accurate upper bound on the support size of the distribution in question. Both the algorithm proposed in this current work and that of [1] employ linear programming, though these programs differ significantly (to the extent that the linear program of [1] does not even have an objective function and simply defines a feasible region). Our proof of the theoretical guarantees in this work leverages some of the machinery of [1] (in particular, the "Chebyshev bump construction") and achieves the same theoretical worst-case optimality guarantees. See Appendix A for further theoretical and practical comparisons with the estimator of [1].

## 1.2 Definitions and examples

We begin by defining the *fingerprint* of a sample, which essentially removes all the label-information from the sample. For the remainder of this paper, we will work with the fingerprint of a sample, rather than the with the sample itself.

**Definition 1.** *Given a samples $X = (x_1, \ldots, x_k)$, the associated* fingerprint, $\mathcal{F} = (\mathcal{F}_1, \mathcal{F}_2, \ldots)$, *is the "histogram of the histogram" of the sample. Formally, $\mathcal{F}$ is the vector whose $i^{th}$ component, $\mathcal{F}_i$, is the number of elements in the domain that occur exactly $i$ times in sample $X$.*

For estimating entropy, or any other property whose value is invariant to relabeling the distribution support, the fingerprint of a sample contains all the relevant information (see [21], for a formal proof of this fact). We note that in some of the literature, the fingerprint is alternately termed the *pattern*, *histogram*, *histogram of the histogram* or *collision statistics* of the sample.

In analogy with the fingerprint of a sample, we define the *histogram* of a distribution, a representation in which the labels of the domain have been removed.

**Definition 2.** *The* histogram *of a distribution $D$ is a mapping $h_D : (0, 1] \to \mathbb{N} \cup \{0\}$, where $h_D(x)$ is equal to the number of domain elements that each occur in distribution $D$ with probability $x$. Formally, $h_D(x) = |\{\alpha : D(\alpha) = x\}|$, where $D(\alpha)$ is the probability mass that distribution $D$ assigns to domain element $\alpha$. We will also allow for "generalized histograms" in which $h_D$ does not necessarily take integral values.*

Since $h(x)$ denotes the number of elements that have probability $x$, we have $\sum_{x:h(x)\neq 0} x \cdot h(x) = 1$, as the total probability mass of a distribution is 1. Any *symmetric* property is a function of only the histogram of the distribution:

- The *Shannon entropy $H(D)$* of a distribution $D$ is defined to be
$$H(D) := -\sum_{\alpha \in sup(D)} D(\alpha) \log_2 D(\alpha) = -\sum_{x:h_D(x)\neq 0} h_D(x) x \log_2 x.$$
- The *support size* is the number of domain elements that occur with positive probability:
$$|sup(D)| := |\{\alpha : D(\alpha) > 0\}| = \sum_{x:h_D(x)\neq 0} h_D(x).$$

We provide an example to illustrate the above definitions:

**Example 3.** *Consider a sequence of animals, obtained as a sample from the distribution of animals on a certain island, $X = (mouse, mouse, bird, cat, mouse, bird, bird, mouse, dog, mouse)$. We have $\mathcal{F} = (2, 0, 1, 0, 1)$, indicating that two species occurred exactly once (cat and dog), one species occurred exactly three times (bird), and one species occurred exactly five times (mouse).*

*Consider the following distribution of animals:*

$$Pr(mouse) = 1/2, \quad Pr(bird) = 1/4, \quad Pr(cat) = Pr(dog) = Pr(bear) = Pr(wolf) = 1/16.$$

*The associated* histogram *of this distribution is $h : (0, 1] \to \mathbb{Z}$ defined by $h(1/16) = 4$, $h(1/4) = 1$, $h(1/2) = 1$, and for all $x \notin \{1/16, 1/4, 1/2\}$, $h(x) = 0$.*

As we will see in Example 5 below, the fingerprint of a sample is intimately related to the Binomial distribution; the theoretical analysis will be greatly simplified by reasoning about the related Poisson distribution, which we now define:

**Definition 4.** *We denote the Poisson distribution of expectation $\lambda$ as $Poi(\lambda)$, and write $poi(\lambda, j) := \frac{e^{-\lambda}\lambda^j}{j!}$, to denote the probability that a random variable with distribution $Poi(\lambda)$ takes value $j$.*

**Example 5.** *Let $D$ be the uniform distribution with support size* $1000$. *Then $h_D(1/1000) = 1000$, and for all $x \neq 1/1000$, $h_D(x) = 0$. Let $X$ be a sample consisting of $500$ independent draws from $D$. Each element of the domain, in expectation, will occur $1/2$ times in $X$, and thus the number of occurrences of each domain element in the sample $X$ will be roughly distributed as $Poi(1/2)$. (The exact distribution will be $Binomial(500, 1/1000)$, though the Poisson distribution is an accurate approximation.) By linearity of expectation, the expected fingerprint satisfies $E[\mathcal{F}_i] \approx 1000 \cdot poi(1/2, i)$. Thus we expect to see roughly $303$ elements once, $76$ elements twice, $13$ elements three times, etc., and in expectation $607$ domain elements will not be seen at all.*

## 2 Estimating the unseen

Given the fingerprint $\mathcal{F}$ of a sample of size $k$, drawn from a distribution with histogram $h$, our high-level approach is to find a histogram $h'$ that has the property that if one were to take $k$ independent draws from a distribution with histogram $h'$, the fingerprint of the resulting sample would be similar to the observed fingerprint $\mathcal{F}$. The hope is then that $h$ and $h'$ will be similar, and, in particular, have similar entropies, support sizes, etc.

As an illustration of this approach, suppose we are given a sample of size $k = 500$, with fingerprint $\mathcal{F} = (301, 78, 13, 1, 0, 0, \ldots)$; recalling Example 5, we recognize that $\mathcal{F}$ is very similar to the expected fingerprint that we would obtain if the sample had been drawn from the uniform distribution over support $1000$. Although the sample only contains $391$ unique domain elements, we might be justified in concluding that the entropy of the true distribution from which the sample was drawn is close to $H(Unif(1000)) = \log_2(1000)$.

In general, how does one obtain a "plausible" histogram from a fingerprint in a principled fashion? We must start by understanding how to obtain a plausible fingerprint from a histogram.

Given a distribution $D$, and some domain element $\alpha$ occurring with probability $x = D(\alpha)$, the probability that it will be drawn exactly $i$ times in $k$ independent draws from $D$ is $Pr[Binomial(k, x) = i] \approx poi(kx, i)$. By linearity of expectation, the expected $i$th fingerprint entry will roughly satisfy

$$E[\mathcal{F}_i] \approx \sum_{x: h_D(x) \neq 0} h(x) poi(kx, i). \tag{1}$$

This mapping between histograms and expected fingerprints is linear in the histogram, with coefficients given by the Poisson probabilities. Additionally, it is not hard to show that $Var[\mathcal{F}_i] \leq E[\mathcal{F}_i]$, and thus the fingerprint is tightly concentrated about its expected value. This motivates a "first moment" approach. We will, roughly, invert the linear map from histograms to expected fingerprint entries, to yield a map from observed fingerprints, to plausible histograms $h'$.

There is one additional component of our approach. For many fingerprints, there will be a large space of equally plausible histograms. To illustrate, suppose we obtain fingerprint $\mathcal{F} = (10, 0, 0, 0, \ldots)$, and consider the two histograms given by the uniform distributions with respective support sizes 10,000, and 100,000. Given either distribution, the probability of obtaining the observed fingerprint from a set of 10 samples is $> .99$, yet these distributions are quite different and have very different entropy values and support sizes. They are both very plausible–which distribution should we return?

To resolve this issue in a principled fashion, we strengthen our initial goal of "returning a histogram that could have plausibly generated the observed fingerprint": we instead return the *simplest* histogram that could have plausibly generated the observed fingerprint. Recall the example above, where we observed only 10 distinct elements, but to explain the data we could either infer an additional 9,900 unseen elements, or an additional 99,000. In this sense, inferring "only" 9,900 additional unseen elements is the simplest explanation that fits the data, in the spirit of Occam's razor.[2]

### 2.1 The algorithm

We pose this problem of finding the simplest plausible histogram as a pair of linear programs. The first linear program will return a histogram $h'$ that minimizes the distance between its expected fingerprint and the observed fingerprint, where we penalize the discrepancy between $\mathcal{F}_i$ and $E[\mathcal{F}_i^{h'}]$ in proportion to the inverse of the standard deviation of $\mathcal{F}_i$, which we estimate as $1/\sqrt{1 + \mathcal{F}_i}$, since

Poisson distributions have variance equal to their expectation. The constraint that $h'$ corresponds to a histogram simply means that the total probability mass is 1, and all probability values are nonnegative. The second linear program will then find the histogram $h''$ of minimal support size, subject to the constraint that the distance between its expected fingerprint, and the observed fingerprint, is not much worse than that of the histogram found by the first linear program.

To make the linear programs finite, we consider a fine mesh of values $x_1, \ldots, x_\ell \in (0, 1]$ that between them discretely approximate the potential support of the histogram. The variables of the linear program, $h'_1, \ldots, h'_\ell$ will correspond to the histogram values at these mesh points, with variable $h'_i$ representing the number of domain elements that occur with probability $x_i$, namely $h'(x_i)$.

A minor complicating issue is that this approach is designed for the challenging "rare events" regime, where there are many domain elements each seen only a handful of times. By contrast if there is a domain element that occurs very frequently, say with probability $1/2$, then the number of times it occurs will be concentrated about its expectation of $k/2$ (and the trivial empirical estimate will be accurate), though fingerprint $\mathcal{F}_{k/2}$ will not be concentrated about its expectation, as it will take an integer value of either $0, 1$ or $2$. Hence we will split the fingerprint into the "easy" and "hard" portions, and use the empirical estimator for the easy portion, and our linear programming approach for the hard portion. The full algorithm is below (see our websites or Appendix D for Matlab code).

---

**Algorithm 1.** ESTIMATE UNSEEN

Input: Fingerprint $\mathcal{F} = \mathcal{F}_1, \mathcal{F}_2, \ldots, \mathcal{F}_m$, derived from a sample of size $k$,
  vector $x = x_1, \ldots, x_\ell$ with $0 < x_i \leq 1$, and error parameter $\alpha > 0$.
Output: List of pairs $(y_1, h'_{y_1}), (y_2, h'_{y_2}), \ldots$, with $y_i \in (0, 1]$, and $h'_{y_i} \geq 0$.

- Initialize the output list of pairs to be empty, and initialize a vector $\mathcal{F}'$ to be equal to $\mathcal{F}$.
- For $i = 1$ to $k$,
  - If $\sum_{j \in \{i - \lceil \sqrt{i} \rceil, \ldots, i + \lceil \sqrt{i} \rceil\}} \mathcal{F}_j \leq 2\sqrt{i}$       [i.e. if the fingerprint is "sparse" at index $i$]
    Set $\mathcal{F}'_i = 0$, and append the pair $(i/k, \mathcal{F}_i)$ to the output list.
- Let $v_{opt}$ be the objective function value returned by running Linear Program 1 on input $\mathcal{F}', x$.
- Let $h$ be the histogram returned by running Linear Program 2 on input $\mathcal{F}', x, v_{opt}, \alpha$.
- For all $i$ s.t. $h_i > 0$, append the pair $(x_i, h_i)$ to the output list.

**Linear Program 1.** FIND PLAUSIBLE HISTOGRAM

Input: Fingerprint $\mathcal{F} = \mathcal{F}_1, \mathcal{F}_2, \ldots, \mathcal{F}_m$, derived from a sample of size $k$,
  vector $x = x_1, \ldots, x_\ell$ consisting of a fine mesh of points in the interval $(0, 1]$.
Output: vector $h' = h'_1, \ldots, h'_\ell$, and objective value $v_{opt} \in \mathbb{R}$.

Let $h'_1, \ldots, h'_\ell$ and $v_{opt}$ be, respectively, the solution assignment, and corresponding objective function value of the solution of the following linear program, with variables $h'_1, \ldots, h'_\ell$:

$$\text{Minimize: } \sum_{i=1}^{m} \frac{1}{\sqrt{1 + \mathcal{F}_i}} \left| \mathcal{F}_i - \sum_{j=1}^{\ell} h'_j \cdot poi(kx_j, i) \right|$$

$$\text{Subject to: } \sum_{j=1}^{\ell} x_j h'_j = \sum_i \mathcal{F}_i / k, \text{ and } \forall j, \ h'_j \geq 0.$$

**Linear Program 2.** FIND SIMPLEST PLAUSIBLE HISTOGRAM

Input: Fingerprint $\mathcal{F} = \mathcal{F}_1, \mathcal{F}_2, \ldots, \mathcal{F}_m$, derived from a sample of size $k$,
  vector $x = x_1, \ldots, x_\ell$ consisting of a fine mesh of points in the interval $(0, 1]$,
  optimal objective function value $v_{opt}$ from Linear Program 1, and error parameter $\alpha > 0$.
Output: vector $h' = h'_1, \ldots, h'_\ell$.
Let $h'_1, \ldots, h'_\ell$ be the solution assignment of the following linear program, with variables $h'_1, \ldots, h'_\ell$:

$$\text{Minimize: } \sum_{j=1}^{\ell} h'_j \qquad \text{Subject to: } \sum_{i=1}^{m} \frac{1}{\sqrt{1 + \mathcal{F}_i}} \left| \mathcal{F}_i - \sum_{j=1}^{\ell} h'_j \cdot poi(kx_j, i) \right| \leq v_{opt} + \alpha,$$

$$\sum_{j=1}^{\ell} x_j h'_j = \sum_i \mathcal{F}_i / k, \text{ and } \forall j, \ h'_j \geq 0.$$

---

**Theorem 1.** *There exists a constant $C_0 > 0$ and assignment of parameter $\alpha := \alpha(k)$ of Algorithm 1 such that for any $c > 0$, for sufficiently large $n$, given a sample of size $k = c\frac{n}{\log n}$ consisting of independent draws from a distribution $D$ over a domain of size at most $n$, with probability at least $1 - e^{-n^{\Omega(1)}}$ over the randomness in the selection of the sample, Algorithm 1[3], when run with a sufficiently fine mesh $x_1, \ldots, x_\ell$, returns a histogram $h'$ such that $|H(D) - H(h')| \leq \frac{C_0}{\sqrt{c}}$.*

The above theorem characterizes the worst-case performance guarantees of the above algorithm in terms of entropy estimation. The proof of Theorem 1 is rather technical and we provide the complete proof together with a high-level overview of the key components, in Appendix C. In fact, we prove a stronger theorem—guaranteeing that the histogram returned by Algorithm 1 is close (in a specific metric) to the histogram of the true distribution; this stronger theorem then implies that Algorithm 1 can accurately estimate *any* statistical property that is sufficiently Lipschitz continuous with respect to the specific metric on histograms.

The information theoretic lower bounds of [1] show that there is some constant $C_1$ such that for sufficiently large $k$, *no* algorithm can estimate the entropy of (worst-case) distributions of support size $n$ to within $\pm 0.1$ with any probability of success greater $0.6$ when given a sample of size at most $k = C_1 \frac{n}{\log n}$. Together with Theorem 1, this establishes the worst-case optimality of Algorithm 1 (to constant factors).

## 3 Empirical results

In this section we demonstrate that Algorithm 1 performs well, in practice. We begin by briefly discussing the five entropy estimators to which we compare our estimator in Figure 1. The first three are standard, and are, perhaps, the most commonly used estimators [15]. We then describe two recently proposed estimators that have been shown to perform well [22].

**The "naive" estimator:** the entropy of the empirical distribution, namely, given a fingerprint $\mathcal{F}$ derived from a set of $k$ samples, $H^{naive}(\mathcal{F}) := -\sum_i \mathcal{F}_i \frac{i}{k} |\log_2 \frac{i}{k}|$.

**The Miller-Madow corrected estimator [23]:** the naive estimator $H^{naive}$ corrected to try to account for the second derivative of the logarithm function, namely $H^{MM}(\mathcal{F}) := H^{naive}(\mathcal{F}) + \frac{(\sum_i \mathcal{F}_i) - 1}{2k}$, though we note that the numerator of the correction term is sometimes replaced by various related quantities, see [24].

**The jackknifed naive estimator [25, 26]:** $H^{JK}(\mathcal{F}) := k \cdot H^{naive}(\mathcal{F}) - \frac{k-1}{k} \sum_{j=1}^{k} H^{naive}(\mathcal{F}^{-j})$, where $\mathcal{F}^{-j}$ is the fingerprint given by removing the contribution of the $j$th sample.

**The coverage adjusted estimator (CAE) [27]:** Chao and Shen proposed the CAE, which is specifically designed to apply to settings in which there is a significant component of the distribution that is unseen, and was shown to perform well in practice in [22].[4] Given a fingerprint $\mathcal{F}$ derived from a set of $k$ samples, let $P_s := 1 - \mathcal{F}_1/k$ be the Good–Turing estimate of the probability mass of the "seen" portion of the distribution [9]. The CAE adjusts the empirical probabilities according to $P_s$, then applies the Horvitz–Thompson estimator for population totals [28] to take into account the probability that the elements were seen. This yields:

$$H^{CAE}(\mathcal{F}) := -\sum_i \mathcal{F}_i \frac{(i/k)P_s \log_2 ((i/k)P_s)}{1 - (1 - (i/k)P_s)^k}.$$

**The *Best Upper Bound* estimator [15]:** The final estimator to which we compare ours is the *Best Upper Bound* (BUB) estimator of Paninski. This estimator is obtained by searching for a minimax linear estimator, with respect to a certain error metric. The linear estimators of [2] can be viewed as a variant of this estimator with provable performance bounds.[5] The BUB estimator requires, as input, an upper bound on the support size of the distribution from which the samples are drawn; if the bound provided is inaccurate, the performance degrades considerably, as was also remarked in [22]. In our experiments, we used Paninski's implementation of the BUB estimator (publicly available on his website), with default parameters. For the distributions with finite support, we gave the true support size as input, and thus we are arguably comparing our estimator to the best–case performance of the BUB estimator.

See Figure 1 for the comparison of Algorithm 1 with these estimators.

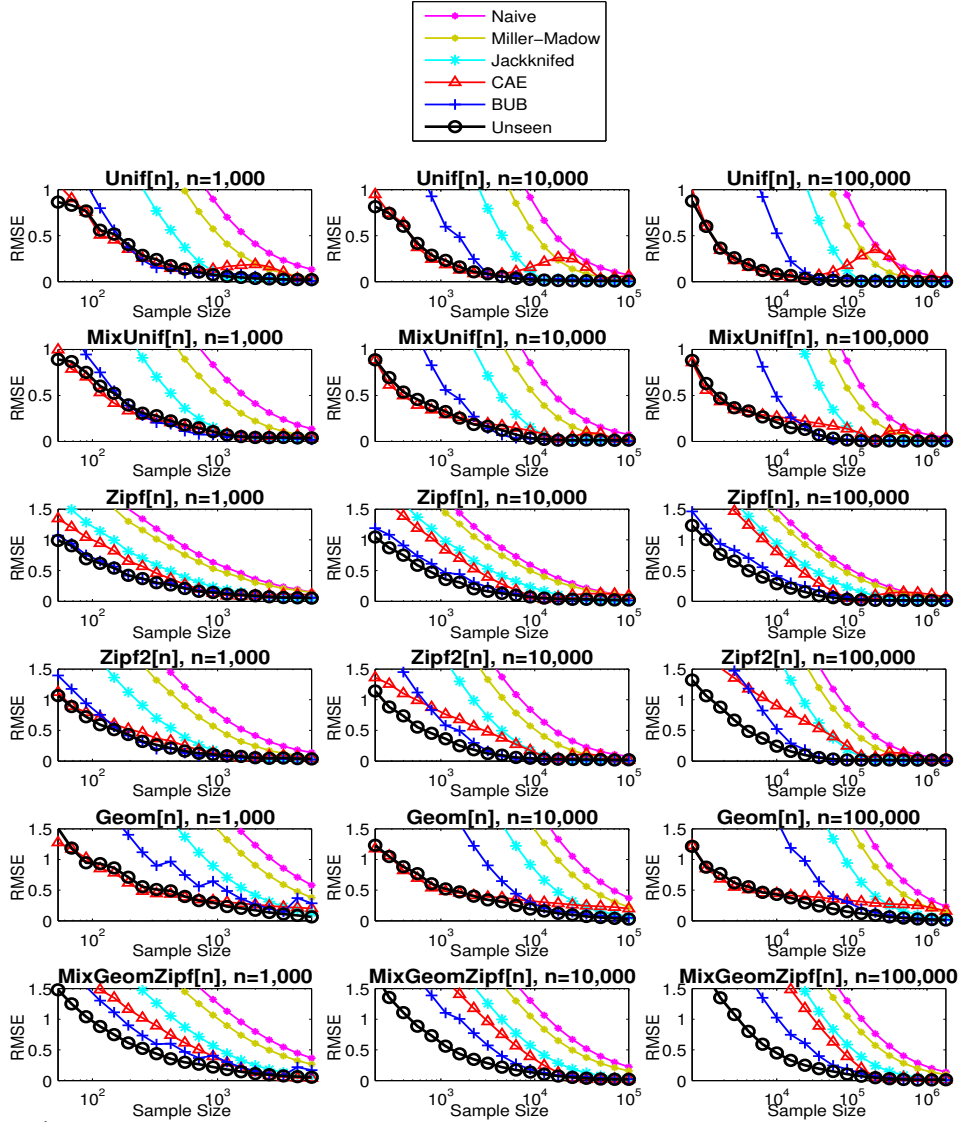

Figure 1: Plots depicting the square root of the mean squared error (RMSE) of each entropy estimator over 500 trials, plotted as a function of the sample size; note the logarithmic scaling of the x-axis. The samples are drawn from six classes of distributions: the uniform distribution, $Unif[n]$ that assigns probability $p_i = 1/n$ for $i = 1, 2, \ldots, n$; an even mixture of $Unif[\frac{n}{5}]$ and $Unif[\frac{4n}{5}]$, which assigns probability $p_i = \frac{5}{2n}$ for $i = 1, \ldots, \frac{n}{5}$ and probability $p_i = \frac{5}{8n}$ for $i = \frac{n}{5} + 1, \ldots, n$; the Zipf distribution $Zipf[n]$ that assigns probability $p_i = \frac{1/i}{\sum_{j=1}^{n} 1/j}$ for $i = 1, 2, \ldots, n$ and is commonly used to model naturally occurring "power law" distributions, particularly in natural language processing; a modified Zipf distribution with power–law exponent 0.6, $Zipf2[n]$, that assigns probability $p_i = \frac{1/i^{0.6}}{\sum_{j=1}^{n} 1/j^{0.6}}$ for $i = 1, 2, \ldots, n$; the geometric distribution $Geom[n]$, which has infinite support and assigns probability $p_i = (1/n)(1 - 1/n)^i$, for $i = 1, 2 \ldots$; and lastly an even mixture of $Geom[n/2]$ and $Zipf[n/2]$. For each distribution, we considered three settings of the parameter $n$: $n = 1,000$ (left column), $n = 10,000$ (center column), and $n = 100,000$ (right column). In each plot, the sample size ranges over the interval $[n^{0.6}, n^{1.25}]$.

All experiments were run in Matlab. The error parameter $\alpha$ in Algorithm 1 was set to be 0.5 for all trials, and the vector $x = x_1, x_2, \ldots$ used as the support of the returned histogram was chosen to be a coarse geometric mesh, with $x_1 = 1/k^2$, and $x_i = 1.1x_{i-1}$. The experimental results are essentially unchanged if the parameter $\alpha$ varied within the range $[0.25, 1]$, or if $x_1$ is decreased, or if the mesh is made more fine (see Appendix B). Appendix D contains our Matlab implementation of Algorithm 1 (also available from our websites).

The *unseen* estimator performs far better than the three standard estimators, dominates the CAE estimator for larger sample sizes and on samples from the Zipf distributions, and also dominates the BUB estimator, even for the uniform and Zipf distributions for which the BUB estimator received the true support sizes as input.

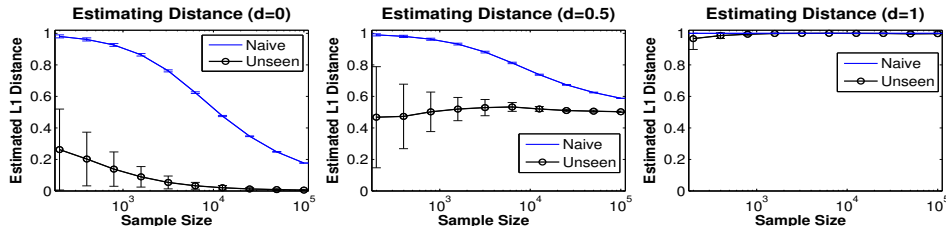

Figure 2: Plots depicting the estimated the total variation distance ($\ell_1$ distance) between two uniform distributions on $n = 10,000$ points, in three cases: the two distributions are identical (left plot, $d = 0$), the supports overlap on *half* their domain elements (center plot, $d = 0.5$), and the distributions have disjoint supports (right plot, $d = 1$). The estimate of the distance is plotted along with error bars at plus and minus one standard deviation; our results are compared with those for the naive estimator (the distance between the empirical distributions). The *unseen* estimator can be seen to reliably distinguish between the $d = 0$, $d = \frac{1}{2}$, and $d = 1$ cases even for samples as small as several hundred.

## 3.1 Estimating $\ell_1$ distance and number of words in *Hamlet*

The other two properties that we consider do not have such widely-accepted estimators as entropy, and thus our evaluation of the unseen estimator will be more qualitative. We include these two examples here because they are of a substantially different flavor from entropy estimation, and highlight the flexibility of our approach.

Figure 2 shows the results of estimating the total variation distance ($\ell_1$ distance). Because total variation distance is a property of two distributions instead of one, fingerprints and histograms are two-dimensional objects in this setting (see Section 4.6 of [29]), and Algorithm 1 and the linear programs are extended accordingly, replacing single indices by pairs of indices, and Poisson coefficients by corresponding products of Poisson coefficients.

Finally, in contrast to the synthetic tests above, we also evaluated our estimator on a real-data problem which may be seen as emblematic of the challenges in a wide gamut of natural language processing problems: *given a (contiguous) fragment of Shakespeare's* Hamlet*, estimate the number of distinct words in the whole play*. We use this example to showcase the flexibility of our linear programming approach—our estimator can be customized to particular domains in powerful and principled ways by adding or modifying the constraints of the linear program. To estimate the histogram of word frequencies in *Hamlet*, we note that the play is of length $\approx 25,000$, and thus the minimum probability with which any word can occur is $\frac{1}{25,000}$. Thus in contrast to our previous approach of using Linear Program 2 to bound the support of the returned histogram, we instead simply modify the input vector $x$ of Linear Program 1 to contain only probability values $\geq \frac{1}{25,000}$, and forgo running Linear Program 2. The results are plotted in Figure 3. The estimates converge towards the true value of 4268 distinct words extremely rapidly, and are slightly negatively biased, perhaps reflecting the fact that words appearing close together are correlated.

In contrast to Hamlet's charge that "there are more things in heaven and earth...than are dreamt of in your philosophy," we can say that there are almost exactly as many things in *Hamlet* as can be dreamt of from 10% of *Hamlet*.

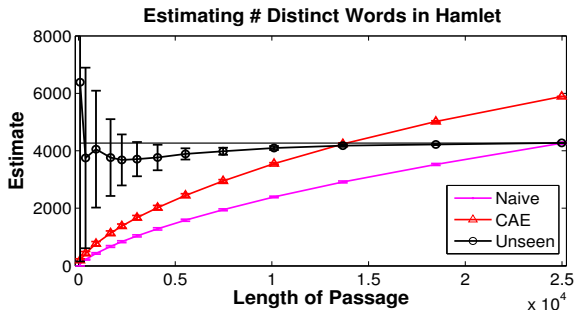

Figure 3: Estimates of the total number of distinct word forms in Shakespeare's *Hamlet* (excluding stage directions and proper nouns) as a functions of the length of the passage from which the estimate is inferred. The true value, 4268, is shown as the horizontal line.

## Footnotes

[1]Three recent studies (appearing in Science last year) found that very rare genetic mutations are especially abundant in humans, and observed that better statistical tools are needed to characterize this "rare events" regime, so as to resolve fundamental problems about our evolutionary process and selective pressures [3, 4, 5].

[2]The practical performance seems virtually unchanged if one returns the "plausible" histogram of minimal entropy, instead of minimal support size (see Appendix B).

[3]For simplicity, we prove this statement for Algorithm 1 with the second bullet step of the algorithm modified as follows: there is an explicit cutoff $N$ such that the linear programming approach is applied to fingerprint entries $\mathcal{F}_i$ for $i \leq N$, and the empirical estimate is applied to fingerprints $\mathcal{F}_i$ for $i > N$.

[4]One curious weakness of the CAE, is that its performance is exceptionally poor on some simple large instances. Given a sample of size $k$ from a uniform distribution over $k$ elements, it is not hard to show that the bias of the CAE is $\Omega(\log k)$. This error is not even bounded! For comparison, even the naive estimator has error bounded by a constant in the limit as $k \to \infty$ in this setting. This bias of the CAE is easily observed in our experiments as the "hump" in the top row of Figure 1.

[5]We also implemented the linear estimators of [2], though found that the BUB estimator performed better.

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
