[Supplementary Material · appendixVV.pdf]

In this Addendum, A) we compare our approach to that of [1] from both a theoretical and practical standpoint; B) we show that the performance of Algorithm 1 is robust to variations and choice of parameters; C) we provide a self-contained proof of (a generalization of) Theorem 1; and D) we include a Matlab implementation of Algorithm 1, which is also available from our websites.

## A    Comparison with [1]

The estimators of [1] differ from the one presented here in several respects. First, they require, as input, an upper bound, $n$, on the true support size of the distribution from which the sample was drawn. Second, rather than adopting the two-stage approach of our estimator, which tries to find the plausible histogram of minimal support size, their approach uses a single linear program, which simply tries to find a plausible histogram. Specifically, their linear program lacks an objective function, and only defines a feasible polytope that consists of all histograms $h'$ whose expected fingerprint is sufficiently close to the observed fingerprint (specifically, $|\operatorname{E}_{h'}[\mathcal{F}_i] - \mathcal{F}_i| \le n^{.51}$). The third difference, which significantly complicates the proof of Theorem 1, is how we quantify "close to the observed fingerprint". Our algorithm measures the distance between the expected fingerprint of a histogram, and the observed fingerprint, by weighting the discrepancy in the $i$th entry by $\frac{1}{\sqrt{\mathcal{F}_i+1}}$. This makes intuitive sense, as the variance in the $i$th fingerprint entry is roughly equal to its expectation (as in a Poisson distribution), and $\mathcal{F}_i$ is a proxy for the expected value of the $i$th fingerprint entry: in short, the objective value of our linear program tries to find a distribution to fit the data so as to minimize the "total error, measured in units of standard deviations". The linear program of [1] simply requires that $|\operatorname{E}_{h'}[\mathcal{F}_i] - \mathcal{F}_i| \le n^{.51}$, irrespective of value of $\mathcal{F}_i$. One of the significant technical hurdles of our proof of Theorem 1 can be roughly viewed as showing that the results of [1] still hold if $n^{.51}$ were instead replaced by $n^{.01}\sqrt{\mathcal{F}_i + 1}$.

In Figure 4 we give empirical evidence for the importance of our two-stage approach—in particular, minimizing the support size while ensuring that the returned histogram still has the property that its expected fingerprints are close to the observed ones.

## B    Robustness to modifying parameters

In this section we give strong empirical evidence for the robustness of our approach. Specifically, we show that the performance of our estimator remains essentially unchanged over large ranges of the two parameters of our estimator: the choice of mesh points of the interval $(0, 1]$ which correspond to the variables of the linear programs, and the parameter $\alpha$ of the second linear program that dictates the additional allowable discrepancy between the expected fingerprints of the returned histogram and the observed fingerprints.

Additionally, we also consider the variant of the second linear program which is based on a slightly different interpretation of Occam's Razor: instead of minimizing the support size of the returned histogram, we now minimize the *entropy* of the returned histogram. Note that this is still a *linear* objective function, and hence can still be solved by a linear program. Formally, recall that the linear programs have variables $h'_1, \ldots, h'_\ell$ corresponding to the histogram values at corresponding fixed grid points $x_1, \ldots, x_\ell$. Rather than having the second linear program minimize $\sum_{j=1}^{\ell} h'_j$, we consider replacing the objective function by

$$\text{Minimize: } \sum_{j=1}^{\ell} h'_j \cdot \log \frac{1}{x_j}.$$

Note that the quantity $\sum_{j=1}^{\ell} h'_j \cdot \log \frac{1}{x_j}$ is precisely the entropy corresponding to the histogram defined by $h(x_i) = h'_i$ and $h(x) = 0$ for all $x \notin \{x_1, \ldots, x_\ell\}$. Additionally, this expression is still a linear function (of the variables $h'_j$) and hence we still have a linear program.

Figure 5 depicts the performance of our estimator with five different sets of parameters, as well as the performance of the estimator with the entropy minimization objective, as described in the previous paragraph.

Figure 4: Comparison between our main algorithm using two linear programs, versus running only the first linear program. Plots depict the square root of the mean squared error (RMSE) of each entropy estimator over 100 trials, plotted as a function of the sample size (note the logarithmic scaling of the x-axis). The samples are drawn from a uniform distribution $Unif[n]$ (top row), a Zipf distribution $Zipf[n]$ (middle row), and a geometric distribution $Geom[n]$ (bottom row), for $n = 1000$ (left column), $n = 10,000$ (middle column), and $n = 100,000$ (right column). Note that the estimator obtained by removing the second linear program (the program that minimizes the support size for "plausible" histograms) performs significantly less consistently than the proposed two-program estimator, and has performance quirks that depend on the distribution family.

Figure 5: Plots depicting the square root of the mean squared error (RMSE) of each entropy estimator over 100 trials, plotted as a function of the sample size. The samples are drawn from a uniform distribution $Unif[n]$ (top row), a Zipf distribution $Zipf[n]$ (middle row), and a geometric distribution $Geom[n]$ (bottom row), for $n = 1000$ (left column), $n = 10,000$ (middle column), and $n = 100,000$ (right column). The unseen estimator with parameters $\alpha, c$ corresponds to setting the error parameter $\alpha$ of Algorithm 1 and the mesh corresponding to the linear program variables to be a geometrically spaced grid with geometric ratio $c$; namely, $X = \left\{ \frac{1}{k^2}, \frac{c}{k^2}, \frac{c^2}{k^2}, \frac{c^3}{k^2}, \dots, \right\}$, where $k$ is the sample size. Note that the performance of the different variants of the *unseen* estimator perform nearly identically. In particular, the performance is essentially unchanged if one makes the granularity of the grid spacing of the mesh of probabilities used in the linear programs more fine, or slightly more coarse. The performance is also essentially identical if one changes the objective function of Linear Program 2 to minimize the entropy of the returned histogram ("Unseen-MinEntropy" in the above plot), rather than minimizing the support size. The performance varies slightly when the error parameter $\alpha$ is changed, though is reasonably robust to increasing or decreasing $\alpha$ by factors of up to 2.

# C   Proof of main theorem

We now give a self-contained proof of Theorem 1. In fact, we will prove a more general theorem that guarantees that Algorithm 1 will, with very high probability, return a histogram which is "close" to the histogram of the true distribution from which the sample was drawn. In particular, for any sufficiently "nice" statistical property of the distribution (such as entropy) that is a function of only the histogram of a distribution, the property value of the histogram returned by our algorithm will be an accurate approximation of the property value of the true distribution from which the sample was drawn.

In order to formally state this more general theorem, we now define what it means for two histograms to be "close".

**Definition 6.** *For two distributions $p_1, p_2$ with respective histograms $h_1, h_2$, we define the* relative earthmover distance *between them, $R(p_1, p_2) := R(h_1, h_2)$, as the minimum over all schemes of moving the probability mass of the first histogram to yield the second histogram, of the cost of moving that mass, where the per-unit mass cost of moving mass from probability $x$ to $y$ is $|\log(x/y)|$. Formally, for $x, y \in (0, 1]$, the cost of moving $x \cdot h(x)$ units of mass from probability $x$ to $y$ is $x \cdot h(x)|\log \frac{x}{y}|$.*

One can also define the relative earthmover distance via the following dual formulation (given by the Kantorovich-Rubinstein theorem, though it can be intuitively seen as exactly what one would expect from linear programming duality):

$$R(h_1, h_2) = \sup_{f \in \mathcal{R}} \sum_{x: h_1(x) + h_2(x) \neq 0} f(x) \cdot x \left( h_1(x) - h_2(x) \right),$$

where $\mathcal{R}$ is the set of differentiable functions $f : (0, 1] \to \mathbb{R}$, s.t. $|\frac{d}{dx} f(x)| \leq \frac{1}{x}$.

We provide a clarifying example of the above definition:

**Example 7.** *Let $p_1 = Unif[m]$, $p_2 = Unif[\ell]$ be the uniform distributions over $m$ and $\ell$ distinct elements, respectively. $R(p_1, p_2) = |\log m - \log \ell|$, since we must take all the probability mass at probability $x = 1/m$ in the histogram corresponding to $p_1$, and move it to probability $y = 1/\ell$, at a per-unit mass cost of $|\log \frac{m}{\ell}| = |\log m - \log \ell|$.*

Throughout, we will restrict our attention to properties that satisfy a weak notion of continuity, defined via the relative earthmover distance.

**Definition 8.** *A symmetric distribution property $\pi$ is $(\epsilon, \delta)$-continuous if for all distributions $p_1, p_2$ with respective histograms $h_1, h_2$ satisfying $R(h_1, h_2) \leq \delta$ it follows that $|\pi(p_1) - \pi(p_2)| \leq \epsilon$.*

We note that both entropy and support size are easily seen to be continuous with respect to the relative earthmover distance.

**Fact 9.** *For a distribution $p$ of support size at most $n$, and $\delta > 0$*

- *The entropy, $H(p) := -\sum_i p(i) \cdot \log p(i)$ is $(\delta, \delta)$-continuous, with respect to the relative earthmover distance.*

- *The support size $S(p) := |\{i : p(i) > 0\}|$ is $(n\delta, \delta)$-continuous, with respect to the relative earthmover distance, over the set of distributions which have no probabilities in the interval $(0, \frac{1}{n})$.*

## C.1   Formal description of algorithm

We now formally state the algorithm to which our theorem applies. The linear program employed by this algorithm is identical to Linear Program 2 (up to renaming variables). The one difference between this algorithm, and Algorithm 1 is the manner in which the fingerprint is partitioned into the "easy" regime for which the empirical estimate is applied, and the "hard" regime for which the linear programming approach is applied. Here, for simplicity, we analyze the partitioning scheme that simply chooses a fixed cutoff, and applies the naive empirical estimator to any fingerprint entry $\mathcal{F}_i$ for $i$ above the cutoff, and applies the linear programming approach to the smaller fingerprint indices.

For clarity of exposition, we state the algorithm in terms of three positive constants, $\mathcal{B}, \mathcal{C},$ and $\mathcal{D}$, which can be defined arbitrarily provided the following inequalities hold:

$$0.1 > \mathcal{B} > \mathcal{C} > \mathcal{B}(\frac{1}{2} + \mathcal{D}) > \frac{\mathcal{B}}{2} > \mathcal{D} > 0.$$

---

**Linear Program 3.**

Given a $k$-sample fingerprint $\mathcal{F}$:

- Define the set $X := \{\frac{1}{k^2}, \frac{2}{k^2}, \frac{3}{k^2}, \ldots, \frac{k^{\mathcal{B}} + k^{\mathcal{C}}}{k}\}$.
- For each $x \in X$, define the associated LP variable $v_x$.

The linear program is defined as follows:

$$\text{Minimize} \sum_{x \in X} v_x, \text{ (minimize support size)}$$

Subject to:

- $\sum_{i=1}^{k^{\mathcal{B}}} \frac{1}{\sqrt{\mathcal{F}_i + 1}} \left| \mathcal{F}_i - \sum_{x \in X} poi(kx, i) v_x \right| \leq k^{2\mathcal{B}}$ (expected fingerprints of $v_x$ are close to $\mathcal{F}$)

- $\sum_{x \in X} x \cdot v_x + \sum_{i=k^{\mathcal{B}} + 2k^{\mathcal{C}}}^{k} \frac{i}{k} \mathcal{F}_i = 1$ (total prob. mass = 1)

- $\forall x \in X, v_x \geq 0$ (histogram entries are non-negative)

---

**Algorithm 2.** ESTIMATE UNSEEN

**Input:** $k$-sample fingerprint $\mathcal{F}$.

**Output:** Generalized histogram $g_{LP}$.

- Let $v = (v_{x_1}, v_{x_2}, \ldots)$ be the solution to Linear Program 3, on input $\mathcal{F}$.
- Let $g_{LP}$ be the generalized histogram formed by setting $g_{LP}(x_i) = v_{x_i}$ for all $i$, and then for each integer $j \geq k^{\mathcal{B}} + 2k^{\mathcal{C}}$, incrementing $g_{LP}(\frac{j}{k})$ by $\mathcal{F}_j$.

---

The following theorem characterizes the performance of the above algorithm. Theorem 1 follows immediately from the following theorem, together with Fact 9 which shows that if two histograms are close in relative earthmover distance, then their entropies are comparably close.

**Theorem 2.** *For any $c > 0$, for sufficiently large $n$, given a sample of size $k = c\frac{n}{\log n}$ consisting of independent draws from a distribution $p \in \mathcal{D}^n$, with probability at least $1 - e^{-n^{\Omega(1)}}$ over the randomness in the selection of the sample, Algorithm 2 returns a generalized histogram $g_{LP}$ such that*

$$R(p, g_{LP}) \leq O\left(\frac{1}{\sqrt{c}}\right).$$

## C.2 Proof approach

The proof of Theorem 2 decomposes into three main parts. The first part of the proof argues that with high probability (over the randomness in the independent draws of the sample) the sample will be a "faithful" sample from the distribution—no domain element occurs too much more frequently than one would expect, and the fingerprint entries are reasonably close to their expected values. This part of the proof, while slightly tedious, follows relatively easily from a series of Chernoff bounds. Having thus compartmentalized the probabilistic component of our theorem, we will then argue that Algorithm 2 will be successful whenever it receives a "faithful" sample as input.

The second component of the proof argues that (provided the sample in question is "faithful"), the histogram of the true distribution, rounded so as to be supported at values in the set $X$ of probabilities

corresponding to the linear program variables, is a feasible point of Linear Program 3. (And has objective function value roughly equal to the true support size, since the rounding will not significantly alter the support size.)

The final component of the proof will then argue that, given *any* two feasible points of Linear Program 3 that both have reasonably small objective function value, they must be close in relative earthmover distance. Since we have already established that the histogram of the true distribution (appropriately rounded) will be feasible, and will have small objective function value, it will follow that the solution output by the linear program (which can only have smaller objective function value), must be close to the histogram of the true distribution. This component of the proof closely follows that of [1], and crucially leverages a similar "Chebyshev Bump" construction, though we provide a slightly simplified proof of the key lemmas here for completeness.

### C.3 A feasible point

The following condition defines what it means for a sample from a distribution to be "faithful":

**Definition 10.** *A sample of size $k$ with fingerprint $\mathcal{F}$, drawn from a distribution $p$ with histogram $h$, is said to be* faithful *if the following conditions hold:*

- *For all $i$,*

$$\left| \mathcal{F}_i - \sum_{x:h(x)\neq 0} h(x) \cdot poi(kx, i) \right| \leq \max\left( \mathcal{F}_i^{\frac{1}{2}+\mathcal{D}}, k^{\mathcal{B}(\frac{1}{2}+\mathcal{D})} \right).$$

- *For all domain elements $i$, letting $p(i)$ denote the true probability of $i$, the number of times $i$ occurs in the sample from $p$ differs from its expectation of $k \cdot p(i)$ by at most*

$$\max\left( (k \cdot p(i))^{\frac{1}{2}+\mathcal{D}}, k^{\mathcal{B}(\frac{1}{2}+\mathcal{D})} \right).$$

The following lemma follows easily from basic tail bounds on Poisson random variables, and Chernoff bounds.

**Lemma 11.** *There is a constant $\gamma > 0$ such that for sufficiently large $k$, a sample of size $k$ consisting of independent draws from a fixed distribution is "faithful" with probability at least $1 - e^{-k^\gamma}$.*

*Proof.* We first analyze the case of a $Poi(k)$-sized sample drawn from a distribution with histogram $h$. Thus

$$\mathrm{E}[\mathcal{F}_i] = \sum_{x:h(x)\neq 0} h(x)poi(kx, i).$$

Additionally, the number of times each domain element occurs is independent of the number of times the other domain elements occur, and thus each fingerprint entry $\mathcal{F}_i$ is the sum of independent random $0/1$ variables, representing whether each domain element occurred exactly $i$ times in the sample (i.e. contributing $1$ towards $\mathcal{F}_i$). By independence, Chernoff bounds apply.

We split the analysis into two cases, according to whether $\mathrm{E}[\mathcal{F}_i] \geq k^\mathcal{B}$. If $\mathrm{E}[\mathcal{F}_i] < k^\mathcal{B}$, we have that $\Pr\left[|\mathcal{F}_i - \mathrm{E}[\mathcal{F}_i]| \geq k^{\mathcal{B}(\frac{1}{2}+\mathcal{D})}\right]$ is monotonically increasing as a function of $\mathrm{E}[\mathcal{F}_i]$, and hence for any $\mathrm{E}[\mathcal{F}_i] < k^\mathcal{B}$, this probability is bounded by considering the case that $\mathrm{E}[\mathcal{F}_i] = k^\mathcal{B}$; in this case, Chernoff bounds yield:

$$\Pr\left[|\mathcal{F}_i - \mathrm{E}[\mathcal{F}_i]| \geq \mathrm{E}[\mathcal{F}_i]^{\frac{1}{2}+\mathcal{D}}\right] \leq 2e^{\left(\frac{1}{\mathrm{E}[\mathcal{F}_i]^{1/2-\mathcal{D}}}\right)^2 \frac{\mathrm{E}[\mathcal{F}_i]}{3}} = 2e^{\frac{\mathrm{E}[\mathcal{F}_i]^{2\mathcal{D}}}{3}} = 2e^{k^{2\mathcal{B}\mathcal{C}}/3}.$$

In the case that $\mathrm{E}[\mathcal{F}_i] \geq k^\mathcal{B}$, we have that $\Pr\left[|\mathcal{F}_i - \mathrm{E}[\mathcal{F}_i]| \geq \mathrm{E}[\mathcal{F}_i]^{\frac{1}{2}+\mathcal{D}}\right]$ is monotonically decreasing as a function of $\mathrm{E}[\mathcal{F}_i]$, and hence this quantity is also bounded by the above Chernoff bound in the case that $\mathrm{E}[\mathcal{F}_i] = k^\mathcal{B}$. A union bound over the first $k$ fingerprints shows that the probability that given a sample (consisting of $Poi(k)$ draws), the probability that any of the fingerprint entries violate the first condition of *faithful* is at most $k \cdot 2e^{-\frac{k^{2\mathcal{B}\mathcal{D}}}{3}} \leq e^{-k^{\Omega(1)}}$.

For the second condition of "faithful", by basic tail bounds for the Poisson distribution, $\Pr[|Poi(x) - x| > x^{\frac{1}{2}+\mathcal{D}}] \leq e^{-x^{\Omega(1)}}$), hence for $x = k \cdot p(i) \geq k^{\mathcal{B}}$, the probability that the number of occurrences of domain element $i$ differs from its expectation of $k \cdot p(i)$ by at least $(k \cdot p(i))^{\frac{1}{2}+\mathcal{D}}$ is bounded by $e^{-k^{\Omega(1)}}$. In the case that $x = k \cdot p(i) < k^{\mathcal{B}}$,

$$\Pr[|Poi(x) - x| > k^{\mathcal{B}(\frac{1}{2}+\mathcal{D})}] \leq \Pr[|Poi(k^{\mathcal{B}}) - k^{\mathcal{B}}| > k^{\mathcal{B}(\frac{1}{2}+\mathcal{D})}] \leq e^{-k^{\Omega(1)}}.$$

Thus we have shown that provided we are considering a sample of size $Poi(k)$, the probability that the conditions hold is at least $1 - e^{-k^{\Omega(1)}}$. To conclude, note that $\Pr[Poi(k) = k] > \frac{1}{3\sqrt{k}}$, and hence the probability that the conditions do not hold for a sample of size exactly $k$ (namely, the probability that they do not hold for a sample of size $Poi(k)$, conditioned on the sample size being exactly $k$), is at most a factor of $3\sqrt{k}$ larger, and hence this probability of failure is still $e^{-k^{\Omega(1)}}$, as desired. $\quad\square$

The following lemma guarantees that, provided the sample is "faithful", the corresponding instance of Linear Program 3 admits a feasible point with small objective function value. Furthermore, there exists at least one such near-optimal point which, when regarded as a histogram, is extremely close to the histogram of the true distribution from which the sample was drawn.

**Lemma 12.** *For sufficiently large $k$, and $n < k^{1+\mathcal{B}/2}$: given a distribution of support size at most $n$ with histogram $h$, and a "faithful" sample of size $k$ with fingerprint $\mathcal{F}$, Linear Program 3 corresponding to $\mathcal{F}$ has a feasible point $v'$ with objective value at most $2n$, such that $v'$ is close to the true histogram $h$ in the following sense:*

$$R(h, h_{v'}) \leq O(k^{\mathcal{C}-\mathcal{B}} + k^{\mathcal{B}(-1/2+\mathcal{D})} \log k) = O(\frac{1}{k^{\Omega(1)}}),$$

*where $h_{v'}$ is the generalized histogram that would be returned by Algorithm 2 if $v'$ were used in place of the solution to the linear program, $v$; namely $h_{v'}$ is obtained from $v'$ by appending the distribution of the empirical fingerprint entries $\mathcal{F}_i$ for $i \geq k^{\mathcal{B}} + 2k^{\mathcal{C}}$.*

Recall that the linear program aims to find distributions that "could reasonably have generated" the observed fingerprint $\mathcal{F}$. Following this intuition, we will show that, provided the sample is faithful, the true distribution, $h$, minimally modified, will in fact be such a feasible point $v'$.

Roughly, $v'$ will be defined by taking the portion of $h$ with probabilities at most $\frac{k^{\mathcal{B}}+k^{\mathcal{C}}}{k}$ and rounding the support of $h$ to the closest multiple of $1/k^2$, so as to be supported at points in the set $X = \{1/k^2, 2/k^2, \ldots\}$. We will then need to adjust the total probability mass accounted for in $v'$ so as to ensure that the second constraint of the linear program is satisfied, namely the total (implicit) probability mass is 1; this adjusting of mass must be accomplished while ensuring that the fingerprint expectations do not change significantly, so as to ensure that the first constraint of the linear program is satisfied.

The objective function value of $v'$ will easily be bounded by $2n$, since we are assuming that the support size of the distribution corresponding to the true histogram, $h$, is bounded by $n$, and the rounding will at most double this value. To argue that $v'$ is a feasible point of the linear program, we note that the mesh $X$ is sufficiently fine so as to guarantee that the rounding of the support of a histogram to probabilities that are integer multiples of $1/k^2$ does not greatly change the expected fingerprints, and hence the expected fingerprint entries associated with $v'$ will be close to those of $h$. Our definition of "faithful" guarantees that all fingerprint entries are close to their expectations, and hence the first condition of the linear program will be satisfied. (Intuitively, the reader should be convinced that there is *some* suitably fine mesh for which rounding issues are benign; there is nothing special about $1/k^2$ except that it simplifies some of the proof.)

To bound the relative earthmover distance between the true histogram $h$ and the histogram $h_{v'}$ associated to $v'$, we first note that the portion of $h_{v'}$ corresponding to probabilities below $\frac{k^{\mathcal{B}}+k^{\mathcal{C}}}{k}$ will be extremely similar to $h$, because it was created from $h$. For probabilities above $\frac{k^{\mathcal{B}}+2k^{\mathcal{C}}}{k}$, $h_{v'}$ and $h$ will be similar because these "frequently-occurring" elements will appear close to their expected number of times, by the second condition of "faithful" and hence the relative earthmover distance between the empirical histogram and the true histogram in this frequently-occurring region

will also be small. Finally, the only remaining region is the relatively narrow intermediate region of probabilities, which is narrow enough so that probability mass can be moved arbitrarily within this intermediate region while incurring minimal relative earthmover cost. The formal proof of Lemma 12 containing the details of this argument is given below.

*Proof of Lemma 12.* We explicitly define $v'$ as a function of the true histogram $h$ and fingerprint of the sample, $\mathcal{F}$, as follows:

1. Define $h'$ such that $h'(x) = h(x)$ for all $x \leq \frac{k^\mathcal{B} + k^\mathcal{C}}{k}$, and $h'(x) = 0$ for all $x > \frac{k^\mathcal{B} + k^\mathcal{C}}{k}$.

2. Initialize $v'$ to be 0, and for each $x \geq 1/k^2$ s.t. $h'(x) \neq 0$ increment $v'_{\bar{x}}$ by $h'(x)$, where $\bar{x} = \max(z \in X : z \leq x)$ is $x$ rounded down to the closest point in the set $X = \{1/k^2, 2/k^2, \ldots\}$.

3. Let $m := \sum_{x \in X} x v'_x + m_\mathcal{F}$, where $m_\mathcal{F} := \sum_{i \geq k^\mathcal{B} + 2k^\mathcal{C}} \frac{i}{k}\mathcal{F}_i$. If $m < 1$, increment $v'_y$ by $(1 - m)/y$, where $y = \frac{k^\mathcal{B} + k^\mathcal{C}}{k}$. Otherwise, if $m \geq 1$, for all $x \in X$ scale $v'_x$ by a factor of $s = \frac{1 - m_\mathcal{F}}{m - m_\mathcal{F}}$, after which the total probability mass $m_\mathcal{F} + \sum_{x \in X} x v'_x$ will be 1.

We first note that the above procedure is well-defined, since $m_\mathcal{F} \leq 1$, and hence, when $m > 1$ and the scaling factor $s$ is applied, $s$ will be positive.

We now argue that $v'$ is a feasible point of the linear program. Note that by construction, the second and third conditions of the linear program are trivially satisfied. We now consider the first condition of the linear program. Note that since $\mathcal{C} > \frac{1}{2}\mathcal{B}$, we have $\sum_{i \leq k^\mathcal{B}} poi(k^\mathcal{B} + k^\mathcal{C}, i) = o(1/k)$, so the fact that we are truncating $h$ at probability $\frac{k^\mathcal{B} + k^\mathcal{C}}{k}$ in the first step in our construction of $v'$, has little effect on the first $k^\mathcal{B}$ "expected fingerprints": specifically, for $i \leq k^\mathcal{B}$,

$$\sum_{x : h(x) \neq 0} (h'(x) - h(x)) \, poi(kx, i) = o(1).$$

Together with the first condition of the definition of faithful, by the triangle inequality, for each $i$,

$$\frac{1}{\sqrt{\mathcal{F}_i + 1}} \left| \mathcal{F}_i - \sum_{x : h'(x) \neq 0} h'(x) poi(kx, i) \right| \leq \max\left( \mathcal{F}_i^\mathcal{D}, k^{\mathcal{B}(\frac{1}{2} + \mathcal{D})} \right) + o(1).$$

We now bound the analyze how the discretization contributes to the first constraint of the linear program. To this end, note that $\left| \frac{d}{dx} poi(kx, i) \right| \leq k$, and since we are discretizing to multiples of $1/k^2$, the discretization alters the contribution of each domain element to each "expected fingerprint" by at most $k/k^2 = 1/k$ (including those domain elements with probability $< 1/k^2$ which are effectively rounded to 0). Thus, since the support size is bounded by $n$, the discretization alters each "expected fingerprint" by at most $n/k$, and thus contributes at most $k^\mathcal{B} \frac{n}{k}$ to the quantity $\sum_{i=1}^{k^\mathcal{B}} \frac{1}{\sqrt{\mathcal{F}_i + 1}} \left| \mathcal{F}_i - \sum_{x \in X} poi(kx, i)v'_x \right|$.

To conclude our analysis of the first condition of the linear program for $v'$, we consider the effect of the final adjustment of probability mass in the construction of $v'$. In the case that $m \leq 1$, where $m$ is the amount of mass in $v'$ before the final adjustment (as defined in the final step in the construction of $v'$), mass is added to $v'_y$, where $y = \frac{k^\mathcal{B} + k^\mathcal{C}}{k}$, and thus since $\sum_{i \leq k^\mathcal{B}} poi(ky, i) = o(1/k)$, this added mass—no matter how much—alters each $\sum_{x \in X} v'_x poi(kx, i)$ by at most $o(1)$.

The case where $m > 1$, and we must scale down the low-frequency portion of the distribution by the quantity $s < 1$, involves a more delicate analysis. We first bound $s$ in such a way that we can leverage the definition of "faithful". Recall that by definition at the start of the third step of the construction of $v'$, we have $s = \frac{1 - m_\mathcal{F}}{m - m_\mathcal{F}} = \frac{\sum_{i < k^\mathcal{B} + 2k^\mathcal{C}} \frac{i}{k}\mathcal{F}_i}{\sum_{x \in X} x v'_x}$. We lowerbound this expression via an upperbound on the denominator, noting that $\sum_{x \in X} x v'_x$ is at most the total probability mass below frequency $\frac{k^\mathcal{B} + k^\mathcal{C}}{k}$ in the true histogram $h$, which by Poisson tail bounds is at most $o(1/k)$ less than the total mass implied by expected fingerprints up to $k^\mathcal{B} + 2k^\mathcal{C}$. Namely, letting $\mathrm{E}[\mathcal{F}_i] =$

$\sum_{x:h(x)\neq 0} h(x) \cdot poi(kx,i)$ be the expected fingerprints of sampling from the true distribution, we have $s \geq \frac{\sum_{i<k^\mathcal{B}+2k^\mathcal{C}} \frac{i}{k}\mathcal{F}_i}{\sum_{i<k^\mathcal{B}+2k^\mathcal{C}} \frac{i}{k}\mathrm{E}[\mathcal{F}_i]} - o(1/k)$.

We bound this expression using the definition of "faithful": for each $i$, we have $\mathrm{E}[\mathcal{F}_i] \leq \mathcal{F}_i + \max\left(\mathcal{F}_i^{\frac{1}{2}+\mathcal{D}}, k^{\mathcal{B}(\frac{1}{2}+\mathcal{D})}\right) \leq \mathcal{F}_i + \mathcal{F}_i^{\frac{1}{2}+\mathcal{D}} + k^{\mathcal{B}(\frac{1}{2}+\mathcal{D})}$. To bound $s$, we must bound the sum of these terms, each scaled by $\frac{i}{k}$. Because $x^{\frac{1}{2}+\mathcal{D}}$ is a concave function, and letting $z := \sum_{i<k^\mathcal{B}+2k^\mathcal{C}} \frac{i}{k} = O(\frac{k^{2\mathcal{B}}}{k})$, Jensen's inequality gives that $\sum_{i<k^\mathcal{B}+2k^\mathcal{C}} \frac{i}{k}\mathcal{F}_i^{\frac{1}{2}+\mathcal{D}} \leq z \left(\frac{1}{z}\sum_{i<k^\mathcal{B}+2k^\mathcal{C}} \frac{i}{k}\mathcal{F}_i\right)^{\frac{1}{2}+\mathcal{D}}$. Thus, defining the mass implied by the low-frequency fingerprints to be $m_S := \sum_{i<k^\mathcal{B}+2k^\mathcal{C}} \frac{i}{k}\mathcal{F}_i$, we bound one over the expression in our bound for $s$ as $\frac{\sum_{i<k^\mathcal{B}+2k^\mathcal{C}} \frac{i}{k}\mathrm{E}[\mathcal{F}_i]}{\sum_{i<k^\mathcal{B}+2k^\mathcal{C}} \frac{i}{k}\mathcal{F}_i} \leq 1 + \left(\frac{z}{m_S}\right)^{\frac{1}{2}-\mathcal{D}} + k^{\mathcal{B}(\frac{1}{2}+\mathcal{D})}\frac{z}{m_S}$. Thus $s$ is at least 1 over this last expression, minus $o(1/k)$, which we bound via the inequality $\frac{1}{1+x} \geq 1 - x$ (for positive $x$) as: $s \geq 1 - O(k^{(2\mathcal{B}-1)(\frac{1}{2}-\mathcal{D})})m_S^{-(\frac{1}{2}-\mathcal{D})} - O(k^{2\mathcal{B}+\mathcal{B}(\frac{1}{2}+\mathcal{D})-1})/m_S$.

Recall that $v'$ is scaled by $s$ at the end of the third step of its construction, and thus to analyze the contribution of this scaling to the first constraint of the linear program, we bound the total quantity which will be scaled in the first constraint function, $\sum_{i=1}^{k^\mathcal{B}} \frac{1}{\sqrt{\mathcal{F}_i+1}}\sum_{x\in X} poi(kx,i)v'_x$ at the beginning of step 3. We make use of the bounds on the first constraint derived above, for each $i$:

$$\frac{1}{\sqrt{\mathcal{F}_i+1}}\left|\mathcal{F}_i - \sum_{x:h'(x)\neq 0} poi(kx,i)v'_x\right| \leq \max\left(\mathcal{F}_i^{\mathcal{D}}, k^{\mathcal{B}(\frac{1}{2}+\mathcal{D})}\right) + \frac{n}{k} + o(1),$$

which can be rearranged to

$$\frac{1}{\sqrt{\mathcal{F}_i+1}}\sum_{x:h'(x)\neq 0} poi(kx,i)v'_x \leq \frac{\mathcal{F}_i}{\sqrt{\mathcal{F}_i+1}} + \max\left(\mathcal{F}_i^{\mathcal{D}}, k^{\mathcal{B}(\frac{1}{2}+\mathcal{D})}\right) + \frac{n}{k} + o(1)$$
$$\leq \sqrt{\mathcal{F}_i} + O(k^{\mathcal{B}(\frac{1}{2}+\mathcal{D})}).$$

The Cauchy–Schwarz inequality yields that $\sum_{i\leq k^\mathcal{B}} \sqrt{\mathcal{F}_i} \leq \sqrt{\sum_{i\leq k^\mathcal{B}} \frac{i}{k}\mathcal{F}_i}\sqrt{\sum_{i\leq k^\mathcal{B}} \frac{k}{i}}$, which is bounded by $\sqrt{m_S}O(\sqrt{k\log k})$.

Thus scaling by $s$ in step 3 modifies the first constraint of the linear program by at most the product of $s-1$ and $\frac{1}{\sqrt{\mathcal{F}_i+1}}\sum_{x:h'(x)\neq 0} poi(kx,i)v'_x$, which we have thus bounded as

$$\min\left(1, O(k^{(2\mathcal{B}-1)(\frac{1}{2}-\mathcal{D})})m_S^{-(\frac{1}{2}-\mathcal{D})} + O(k^{2\mathcal{B}+\mathcal{B}(\frac{1}{2}+\mathcal{D})-1})/m_S\right)\left(\sqrt{m_S}O(\sqrt{k\log k}) + O(k^{\mathcal{B}(\frac{3}{2}+\mathcal{D})})\right).$$

When $m_S < k^{3\mathcal{B}-1}$, we bound the left parenthetical expression by 1 and the right expression is bounded by $O(\sqrt{k^{3\mathcal{B}}\log k} + k^{\mathcal{B}(\frac{3}{2}+\mathcal{D})}) = O(k^{\mathcal{B}(\frac{3}{2}+\mathcal{D})})$.

Otherwise, when $m_S \in [k^{3\mathcal{B}-1}, 1]$, we bound the product of the first parenthetical with the rightmost term $O(k^{\mathcal{B}(\frac{3}{2}+\mathcal{D})})$ by simply $O(k^{\mathcal{B}(\frac{3}{2}+\mathcal{D})})$. We bound the remaining two cross-terms as $O(k^{(2\mathcal{B}-1)(\frac{1}{2}-\mathcal{D})})m_S^{-(\frac{1}{2}-\mathcal{D})}\sqrt{m_S}O(\sqrt{k\log k}) \leq O(k^{\mathcal{B}+\mathcal{D}})$ and $O(k^{2\mathcal{B}+\mathcal{B}(\frac{1}{2}+\mathcal{D})-1})/m_S\sqrt{m_S}O(\sqrt{k\log k}) \leq O(k^{\mathcal{B}(1+\mathcal{D})})$. Thus the total contribution of the scaling by $s$ to the first constraint is $O(k^{\mathcal{B}(\frac{3}{2}+\mathcal{D})})$.

Thus for large enough $k$, the first constraint will always be less than $k^{2\mathcal{B}}$

We now turn to analyzing the relative earthmover distance $R(h, h_{v'})$. Consider applying the following earthmoving scheme to $h_{v'}$ to yield a new generalized histogram $g$. The following scheme applies in the case that no probability mass was scaled down from $v'$ in the final step of its construction; in the case that $v'$ was scaled down, we consider applying the same earthmoving scheme, with the modification that one never moves more than $xh_{v'}(x)$ mass from location $x$.

- For each $x \leq \frac{k^{\mathcal{B}}+k^{\mathcal{C}}}{k}$, move $\bar{x}h(x)$ units of probability from location $\bar{x}$ to $x$, where as above, $\bar{x} = \max(z \in X : z \leq x)$ is $x$ rounded down to the closest point in set $X = \{1/k^2, 2/k^2, \ldots\}$.

- For each domain element $i$ that occurs $j \geq k^{\mathcal{B}} + 2k^{\mathcal{C}}$ times, move $\frac{j}{k}$ units of probability mass from location $\frac{j}{k}$ to location $p(i)$, where $p(i)$ is the true probability of domain element $i$.

By our construction of $h_{v'}$, it follows that the above earthmoving scheme is a valid scheme to apply to $h_{v'}$, in the sense that it never tries to move more mass from a point than was at that point. And $g$ is the generalized histogram resulting from applying this scheme to $h_{v'}$. We first show that $R(h_{v'}, g)$ is small, since probability mass is only moved relatively small distances. We will then argue that $R(g, h)$ is small: roughly, this follows from first noting that $g$ and $h$ will be very similar below probability value $\frac{k^{\mathcal{B}}+k^{\mathcal{C}}}{k}$, and from the second condition of "faithful" $g$ and $h$ will also be quite similar above probability $\frac{k^{\mathcal{B}}+4k^{\mathcal{B}}}{k}$. Thus the bulk of the disparity between $g$ and $h$ is in the very narrow intermediate region, within which mass may be moved at the small per-unit-mass cost of $\log \frac{k^{\mathcal{B}}+O(k^{\mathcal{C}})}{k^{\mathcal{B}}} \leq O(k^{\mathcal{C}-\mathcal{B}})$.

We first seek to bound $R(h_{v'}, g)$. To bound the cost of the first component of the scheme, consider some $x \geq \frac{k^{1/2}}{k^2}$. The per-unit-mass cost of applying the scheme at location $x$ is bounded by $\log \frac{x}{x-1/k^2} < 2k^{-1/2}$. From the bound on the support size of $h$ and the construction of $h_{v'}$, the total probability mass in $h_{v'}$ at probabilities $x \leq \frac{k^{1/2}}{k^2}$ is at most $\frac{n}{k^{3/2}} < k^{\mathcal{B}/2-1/2}$, and hence this mass can be moved anywhere at cost $k^{\mathcal{B}/2-1/2} \log(k^2)$. To bound the second component of the scheme, by the second condition of "faithful" for each of these frequently-occurring domain elements that occur $j \geq k^{\mathcal{B}} + 2k^{\mathcal{C}}$ times with true probability $p(i)$, we have that $|k \cdot p(i) - j| \leq (k \cdot p(i))^{\frac{1}{2}+\mathcal{D}}$, and hence the per-unit-mass cost of this portion of the scheme is bounded by $\log \frac{k^{\mathcal{B}} - k^{\mathcal{B}(\frac{1}{2}+\mathcal{D})}}{k^{\mathcal{B}}} \leq O(k^{\mathcal{B}(-\frac{1}{2}+\mathcal{D})})$, which dominates the cost of the first portion of the scheme. Hence

$$R(h_{v'}, g) \leq O(k^{\mathcal{B}(-\frac{1}{2}+\mathcal{D})}).$$

We now consider $R(h, g)$. To this end, we will show that

$$\sum_{x \notin [k^{\mathcal{B}-1}, \frac{k^{\mathcal{B}}+4k^{\mathcal{C}}}{k}]} x|h(x) - g(x)| \leq O(k^{\mathcal{B}(-1/2+\mathcal{D})}).$$

First, consider the case that there was no scaling down of $v'$ in the final step of the construction. For $x \leq k^{\mathcal{B}-1}$, we have $g(x) = \frac{\bar{x}}{x}h(x)$, and hence for $x > \frac{k^{1/2}}{k^2}$, $|h(x) - g(x)| \leq h(x)k^{-1/2}$. On the other hand, $\sum_{x \leq \frac{k^{1/2}}{k^2}} xh(x) \leq k^{-1/2+\mathcal{B}/2}$, since the support size of $h$ is at most $n \leq k^{1+\mathcal{B}/2}$. Including the possible removal of at most $k^{-1/2+\mathcal{D}}$ units of mass during the scaling in the final step of constructing $v'$, we have that

$$\sum_{x \leq k^{\mathcal{B}-1}} x|h(x) - g(x)| \leq O(k^{-1/2+\mathcal{B}/2}).$$

We now consider the "high probability" regime. From the second condition of "faithful", for each domain element $i$ whose true probability is $p(i) \geq \frac{k^{\mathcal{B}}+4k^{\mathcal{C}}}{k}$, the number of times $i$ occurs in the faithful sample will differ from its expectation $k \cdot p(i)$ by at most $(k \cdot p(i))^{\frac{1}{2}+\mathcal{D}}$. Hence from our condition that $\mathcal{C} > \mathcal{B}(\frac{1}{2} + \mathcal{D})$ this element will occur at least $k^{\mathcal{B}} + 2k^{\mathcal{C}}$ times, in which case it will contribute to the portion of $h_{v'}$ corresponding to the empirical distribution. Thus for each such domain element, the contribution to the discrepancy $|h(x) - g(x)|$ is at most $(k \cdot p(i))^{-1/2+\mathcal{D}}$. Hence $\sum_{x \geq k^{\mathcal{B}-1}+4k^{\mathcal{C}-1}} x|h(x) - g(x)| \leq k^{\mathcal{B}(-1/2+\mathcal{D})}$, yielding the claim that

$$\sum_{x \notin [k^{\mathcal{B}-1}, \frac{k^{\mathcal{B}}+4k^{\mathcal{C}}}{k}]} x|h(x) - g(x)| \leq O(k^{\mathcal{B}(-1/2+\mathcal{D})}).$$

To conclude, note that all the probability mass in $g$ and $h$ at probabilities below $1/k^2$ can be moved to location $1/k^2$ incurring a relative earthmover cost bounded by $\max_{x \leq 1/k^2} nx |\log xk^2| \leq \frac{n}{k^2} \leq \frac{k^{\mathcal{B}/2}}{k}$. After such a move, the remaining discrepancy between $g(x)$ and $h(x)$ for $x \notin [\frac{k^{\mathcal{B}}}{k}, \frac{k^{\mathcal{B}}+4k^{\mathcal{C}}}{k}]$ can be moved to probability $k^{\mathcal{B}}/k$ at a per-unit-mass cost of at most $\log k^2$, and hence a total cost of at most $O(k^{\mathcal{B}(-1/2+\mathcal{D})} \log k^2)$. After this move, the only region for which $g(x)$ and $h(x)$ differ is the narrow region with $x \in [\frac{k^{\mathcal{B}}}{k}, \frac{k^{\mathcal{B}}+4k^{\mathcal{C}}}{k}]$, within which mass may be moved arbitrarily at a total cost of $\log(1 + 4k^{\mathcal{C}-\mathcal{B}}) \leq O(k^{\mathcal{C}-\mathcal{B}})$. Hence we have

$$R(h, h_{v'}) \leq R(h, g) + R(g, h_{v'}) \leq O(k^{\mathcal{C}-\mathcal{B}} + k^{\mathcal{B}(-1/2+\mathcal{D})} \log k).$$

$\square$

## C.4 Similar expected fingerprints imply similar histograms

In this section we argue that if two histograms $h_1, h_2$ corresponding to distributions with support size at most $O(n)$ have the property that their expected fingerprints derived from $Poi(k)$-sized samples are very similar, then $R(h_1, h_2)$ must be small. This will guarantee that any two feasible points of Linear Program 3 that both have small objective function values correspond to histograms that are close in relative earthmover distance. The previous section established the existence of a feasible point with small objective function value that is close to the true histogram, hence by the triangle inequality, all such feasible points must be close to the true histogram; in particular, the optimal point—the solution to the linear program—will correspond to a histogram that is close to the true histogram of the distribution from which the sample was drawn, completing our proof of Theorem 2.

We define a class of earthmoving schemes, which will allow us to directly relate the relative earth-mover cost of two distributions to the discrepancy in their respective fingerprint expectations. The main technical tool is a Chebyshev polynomial construction, though for clarity, we first describe a simpler scheme that provides some intuition for the Chebyshev construction. We begin by describing the form of our earthmoving schemes; since we hope to relate the cost of such schemes to the discrepancy in expected fingerprints of $Poi(k)$-sized samples, we will require that the schemes be formulated in terms of the Poisson functions $poi(kx, i)$.

**Definition 13.** *For a given $k$, a $\beta$-bump earthmoving scheme is defined by a sequence of positive real numbers $\{c_i\}$, the bump centers, and a sequence of functions $\{f_i\} : (0, 1] \rightarrow \mathbb{R}$ such that $\sum_{i=0}^{\infty} f_i(x) = 1$ for each $x$, and each function $f_i$ may be expressed as a linear combination of Poisson functions, $f_i(x) = \sum_{j=0}^{\infty} a_{ij} poi(kx, j)$, such that $\sum_{j=0}^{\infty} |a_{ij}| \leq \beta$.*

*Given a generalized histogram $h$, the scheme works as follows: for each $x$ such that $h(x) \neq 0$, and each integer $i \geq 0$, move $xh(x) \cdot f_i(x)$ units of probability mass from $x$ to $c_i$. We denote the histogram resulting from this scheme by $(c, f)(h)$.*

**Definition 14.** *A bump earthmoving scheme $(c, f)$ is $[\epsilon, n]$-good if for any generalized histogram $h$ of support size $\sum_x h(x) \leq n$, the relative earthmover distance between $h$ and $(c, f)(h)$ is at most $\epsilon$.*

The crux of the proof of correctness of our estimator is the explicit construction of a surprisingly good earthmoving scheme. We will show that for any $k$ and $n = \delta k \log k$ for some $\delta \in [1/\log k, 1]$, there exists an $[O(\sqrt{\delta}), n]$-good $O(k^{0.3})$-bump earthmoving scheme. In fact, we will construct a single scheme for all $\delta$. We begin by defining a simple scheme that illustrates the key properties of a bump earthmoving scheme, and its analysis.

Perhaps the most natural bump earthmoving scheme is where the bump functions $f_i(x) = poi(kx, i)$ and the bump centers $c_i = \frac{i}{k}$. For $i = 0$, we may, for example, set $c_0 = \frac{1}{2k}$ so as to avoid a logarithm of 0 when evaluating relative earthmover distance. This is a valid earthmoving scheme since $\sum_{i=0}^{\infty} f_i(x) = 1$ for any $x$.

The motivation for this construction is the fact that, for any $i$, the amount of probability mass that ends up at $c_i$ in $(c, f)(h)$ is exactly $\frac{i+1}{k}$ times the expectation of the $i + 1$st fingerprint in a $Poi(k)$-

sample from $h$:

$$((c,f)(h))(c_i) = \sum_{x:h(x)\neq 0} h(x)x \cdot f_i(x) \quad = \sum_{x:h(x)\neq 0} h(x)x \cdot poi(kx,i)$$

$$= \sum_{x:h(x)\neq 0} h(x) \cdot poi(kx,i+1)\frac{i+1}{k}$$

$$= \frac{i+1}{k}\sum_{x:h(x)\neq 0} h(x) \cdot poi(kx,i+1).$$

Consider applying this earthmoving scheme to two histograms $h,g$ with nearly identical finger-print expectations. Letting $h' = (c,f)(h)$ and $g' = (c,f)(g)$, by definition both $h'$ and $g'$ are supported at the bump centers $c_i$, and by the above equation, for each $i$, $|h'(c_i) - g'(c_i)| = \frac{i+1}{k}|\sum_x (h(x)-g(x))poi(kx,i+1)|$, where this expression is exactly $\frac{i+1}{k}$ times the difference between the $i + 1$st fingerprint expectations of $h$ and $g$. In particular, if $h$ and $g$ have nearly identical fingerprint expectations, then $h'$ and $g'$ will be very similar. Analogs of this relation between $R((c,f)(g),(c,f)(h))$ and the discrepancy between the expected fingerprint entries corresponding to $g$ and $h$ will hold for any bump earthmoving scheme, $(c,f)$. Sufficiently "good" earthmoving schemes (guaranteeing that $R(h,h')$ and $R(g,g')$ are small) thus provides a powerful way of bounding the relative earthmover distance between two distributions in terms of the discrepancy in their fingerprint expectations.

The problem with the "Poisson bump" earthmoving scheme described in the previous paragraph is that it not very "good": it incurs a very large relative earthmover cost, particularly for small probabilities. This is due to the fact that most of the mass that starts at a probability below $\frac{1}{k}$ will end up in the zeroth bump, no matter if it has probability nearly $\frac{1}{k}$, or the rather lower $\frac{1}{n}$. Phrased differently, the problem with this scheme is that the first few "bumps" are extremely fat. The situation gets significantly better for higher Poisson functions: most of the mass of $Poi(i)$ lies within relative distance $O(\frac{1}{\sqrt{i}})$ of $i$, and hence the scheme is relatively cheap for larger probabilities $x \gg \frac{1}{k}$. We will therefore construct a scheme that uses regular Poisson functions $poi(kx,i)$ for $i \geq O(\log k)$, but takes great care to construct "skinnier" bumps below this region.

The main tool of this construction of skinnier bumps is the Chebyshev polynomials. For each integer $i \geq 0$, the $i$th Chebyshev polynomial, denoted $T_i(x)$, is the polynomial of degree $i$ such that $T_i(\cos(y)) = \cos(i \cdot y)$. Thus, up to a change of variables, any linear combination of cosine functions up to frequency $s$ may be re-expressed as the same linear combination of the Chebyshev polynomials of orders 0 through $s$. Given this, constructing a "good" earth-moving scheme is an exercise in trigonometric constructions.

Before formally defining our bump earthmoving scheme, we give a rough sketch of the key features. We define the scheme with respect to a parameter $s = O(\log k)$. For $i > s$, we use the fat Poisson bumps: that is, we define the bump centers $c_i = \frac{i}{k}$ and functions $f_i = poi(kx,i)$. For $i \leq s$, we will use skinnier "Chebyshev bumps"; these bumps will have roughly quadratically spaced bump centers $c_i \approx \frac{i^2}{k\log k}$, with the width of the $i$th bump roughly $\frac{i}{k\log k}$ (as compared to the larger width of $\frac{\sqrt{i}}{k}$ of the $i$th Poisson bump). At a high level, the logarithmic factor improvement in our $O(\frac{n}{\log n})$ bound on the sample size necessary to achieve accurate estimation arises because the first few Chebyshev bumps have width $O(\frac{1}{k\log k})$, in contrast to the first Poisson bump, $poi(kx,1)$, which has width $O(\frac{1}{k})$.

**Definition 15.** *The* Chebyshev bumps *are defined in terms of $k$ as follows. Let $s = 0.2\log k$. Define $g_1(y) = \sum_{j=-s}^{s-1}\cos(jy)$. Define*

$$g_2(y) = \frac{1}{16s}\left(g_1(y - \frac{3\pi}{2s}) + 3g_1(y - \frac{\pi}{2s}) + 3g_1(y + \frac{\pi}{2s}) + g_1(y + \frac{3\pi}{2s})\right),$$

*and, for $i \in \{1,\ldots,s-1\}$ define $g_3^i(y) := g_2(y - \frac{i\pi}{s}) + g_2(y + \frac{i\pi}{s})$, and $g_3^0 = g_2(y)$, and $g_3^s = g_2(y+\pi)$. Let $t_i(x)$ be the linear combination of Chebyshev polynomials so that $t_i(\cos(y)) = g_3^i(y)$. We thus define $s + 1$ functions, the "skinny bumps", to be $B_i(x) = t_i(1 - \frac{xk}{2s})\sum_{j=0}^{s-1}poi(xk,j)$, for $i \in \{0,\ldots,s\}$. That is, $B_i(x)$ is related to $g_3^i(y)$ by the coordinate transformation $x = \frac{2s}{k}(1 - \cos(y))$, and scaling by $\sum_{j=0}^{s-1}poi(xk,j)$.*

The Chebyshev bumps of Definition 15 are "third order"; if, instead, we had considered the analogous less skinny "second order" bumps by defining $g_2(y) :=$ $\frac{1}{8s}\left(g_1(y - \frac{\pi}{s}) + 2g_1(y) + g_1(y + \frac{\pi}{s})\right)$, then the results would still hold, though the proofs are slightly more cumbersome.

**Definition 16.** *The* Chebyshev earthmoving scheme *is defined in terms of $k$ as follows: as in Definition 15, let $s = 0.2 \log k$. For $i \geq s + 1$, define the $i$th bump function $f_i(x) = poi(kx, i - 1)$ and associated bump center $c_i = \frac{i-1}{k}$. For $i \in \{0, \ldots, s\}$ let $f_i(x) = B_i(x)$, and for $i \in \{1, \ldots, s\}$, define their associated bump centers $c_i = \frac{2s}{k}(1 - \cos(\frac{i\pi}{s}))$, and let $c_0 := c_1$.*

The following lemma characterizes the key properties of the Chebyshev earthmoving scheme. Namely, that the scheme is, in fact, an earthmoving scheme, that each bump can be expressed as a low-eight linear combination of Poisson functions, and that the scheme incurs a small relative-earthmover cost.

**Lemma 17.** *The Chebyshev earthmoving scheme, of Definition 16 has the following properties:*

- *For any $x \geq 0$,*

$$\sum_{i \geq 0} f_i(x) = 1,$$

  *hence the Chebyshev earthmoving scheme is a valid earthmoving scheme.*

- *Each $B_i(x)$ may be expressed as $\sum_{j=0}^{\infty} a_{ij} poi(kx, j)$ for $a_{ij}$ satisfying*

$$\sum_{j=0}^{\infty} |a_{ij}| \leq 2k^{0.3}.$$

- *The Chebyshev earthmoving scheme is $[O(\sqrt{\delta}), n]$-good, for $n = \delta k \log k$, and $\delta \geq \frac{1}{\log k}$.*

The proof of the above lemma is quite involved, and we split its proof into a series of lemmas. The first lemma below shows that the Chebyshev scheme is a valid earthmoving scheme (the first bullet in the above lemma):

**Lemma 18.** *For any $x$*

$$\sum_{i=-s+1}^{s} g_2(x + \frac{\pi i}{s}) = 1, \text{ and } \sum_{i=0}^{\infty} f_i(x) = 1.$$

*Proof.* $g_2(y)$ is a linear combination of cosines at integer frequencies $j$, for $j = 0, \ldots, s$, shifted by $\pm \pi/2s$ and $\pm 3\pi/s2$. Since $\sum_{i=-s+1}^{s} g_2(x + \frac{\pi i}{s})$ sums these cosines over all possible multiples of $\pi/s$, we note that all but the frequency 0 terms will cancel. The $\cos(0y) = 1$ term will show up once in each $g_1$ term, and thus $1 + 3 + 3 + 1 = 8$ times in each $g_2$ term, and thus $8 \cdot 2s$ times in the sum in question. Together with the normalizing factor of $16s$, the total sum is thus 1, as claimed.

For the second part of the claim,

$$\sum_{i=0}^{\infty} f_i(x) = \left(\sum_{j=-s+1}^{s} g_2(\cos^{-1}\left(\frac{xk}{2s} - 1\right) + \frac{\pi j}{s})\right) \sum_{j=0}^{s-1} poi(xk, j) + \sum_{j \geq s} poi(xk, j)$$

$$= 1 \cdot \sum_{j=0}^{s-1} poi(xk, j) + \sum_{j \geq s} poi(xk, j) = 1.$$

□

We now show that each Chebyshev bump may be expressed as a low-weight linear combination of Poisson functions.

**Lemma 19.** *Each $B_i(x)$ may be expressed as $\sum_{j=0}^{\infty} a_{ij} poi(kx, j)$ for $a_{ij}$ satisfying*

$$\sum_{j=0}^{\infty} |a_{ij}| \le 2k^{0.3}.$$

*Proof.* Consider decomposing $g_3^i(y)$ into a linear combination of $\cos(\ell y)$, for $\ell \in \{0, \ldots, s\}$. Since $\cos(-\ell y) = \cos(\ell y)$, $g_1(y)$ consists of one copy of $\cos(sy)$, two copies of $\cos(\ell y)$ for each $\ell$ between 0 and $s$, and one copy of $\cos(0y)$; $g_2(y)$ consists of ($\frac{1}{16s}$ times) 8 copies of different $g_1(y)$'s, with some shifted so as to introduce sine components, but these sine components are canceled out in the formation of $g_3^i(y)$, which is a symmetric function for each $i$. Thus since each $g_3$ contains at most two $g_2$'s, each $g_3^i(y)$ may be regarded as a linear combination $\sum_{\ell=0}^{s} \cos(\ell y) b_{i\ell}$ with the coefficients bounded as $|b_{i\ell}| \le \frac{2}{s}$.

Since $t_i$ was defined so that $t_i(\cos(y)) = g_3^i(y) = \sum_{\ell=0}^{s} \cos(\ell y) b_{i\ell}$, by the definition of Chebyshev polynomials we have $t_i(z) = \sum_{\ell=0}^{s} T_\ell(z) b_{i\ell}$. Thus the bumps are expressed as $B_i(x) = \left(\sum_{\ell=0}^{s} T_\ell(1 - \frac{xk}{2s}) b_{i\ell}\right)\left(\sum_{j=0}^{s-1} poi(xk, j)\right)$. We further express each Chebyshev polynomial via its coefficients as $T_\ell(1 - \frac{xk}{2s}) = \sum_{m=0}^{\ell} \beta_{\ell m}(1 - \frac{xk}{2s})^m$ and then expand each term via binomial expansion as $(1 - \frac{xk}{2s})^m = \sum_{q=0}^{m}(-\frac{xk}{2s})^q \binom{m}{q}$ to yield

$$B_i(x) = \sum_{\ell=0}^{s}\sum_{m=0}^{\ell}\sum_{q=0}^{m}\sum_{j=0}^{s-1} \beta_{\ell m}\left(-\frac{xk}{2s}\right)^q \binom{m}{q} b_{i\ell}\, poi(xk, j).$$

We note that in general we can reexpress $x^q\, poi(xk, j) = x^q \frac{x^j k^j e^{-xk}}{j!} = poi(xk, j+q)\frac{(j+q)!}{j! k^q}$, which finally lets us express $B_i$ as a linear combination of Poisson functions, for all $i \in \{0, \ldots, s\}$:

$$B_i(x) = \sum_{\ell=0}^{s}\sum_{m=0}^{\ell}\sum_{q=0}^{m}\sum_{j=0}^{s-1} \beta_{\ell m}\left(-\frac{1}{2s}\right)^q \binom{m}{q}\frac{(j+q)!}{j!} b_{i\ell}\, poi(xk, j+q).$$

It remains to bound the sum of the absolute values of the coefficients of the Poisson functions. That is, by the triangle inequality, it is sufficient to show that

$$\sum_{\ell=0}^{s}\sum_{m=0}^{\ell}\sum_{q=0}^{m}\sum_{j=0}^{s-1} \left| \beta_{\ell m}\left(-\frac{1}{2s}\right)^q \binom{m}{q}\frac{(j+q)!}{j!} b_{i\ell} \right| \le 2k^{0.3}$$

We take the sum over $j$ first: the general fact that $\sum_{m=0}^{\ell}\binom{m+i}{i} = \binom{i+\ell+1}{i+1}$ implies that $\sum_{j=0}^{s-1}\frac{(j+q)!}{j!} = \sum_{j=0}^{s-1}\binom{j+q}{q}q! = q!\binom{s+q}{q+1} = \frac{1}{q+1}\frac{(s+q)!}{(s-1)!}$, and further, since $q \le m \le \ell \le s$ we have $s + q \le 2s$ which implies that this final expression is bounded as $\frac{1}{q+1}\frac{(s+q)!}{(s-1)!} = s\frac{1}{q+1}\frac{(s+q)!}{s!} \le s \cdot (2s)^q$. Thus we have

$$\sum_{\ell=0}^{s}\sum_{m=0}^{\ell}\sum_{q=0}^{m}\sum_{j=0}^{s-1} \left| \beta_{\ell m}\left(-\frac{1}{2s}\right)^q \binom{m}{q}\frac{(j+q)!}{j!} b_{i\ell} \right| \le \sum_{\ell=0}^{s}\sum_{m=0}^{\ell}\sum_{q=0}^{m} \left| \beta_{\ell m} s \binom{m}{q} b_{i\ell} \right|$$

$$= s\sum_{\ell=0}^{s} |b_{i\ell}| \sum_{m=0}^{\ell} |\beta_{\ell m}| 2^m$$

Chebyshev polynomials have coefficients whose signs repeat in the pattern $(+, 0, -, 0)$, and thus we can evaluate the innermost sum exactly as $|T_\ell(2\mathbf{i})|$, for $\mathbf{i} = \sqrt{-1}$. Since we bounded $|b_{i\ell}| \le \frac{2}{s}$ above, the quantity to be bounded is now $s\sum_{\ell=0}^{s}\frac{2}{s}|T_\ell(2\mathbf{i})|$. Since the explicit expression for Chebyshev polynomials yields $|T_\ell(2\mathbf{i})| = \frac{1}{2}\left[(2 - \sqrt{5})^\ell + (2 + \sqrt{5})^\ell\right]$ and since $|2 - \sqrt{5}|^\ell = (2 + \sqrt{5})^{-\ell}$ we finally bound $s\sum_{\ell=0}^{s}\frac{2}{s}|T_\ell(2\mathbf{i})| \le 1 + \sum_{\ell=-s}^{s}(2+\sqrt{5})^\ell < 1 + \frac{2+\sqrt{5}}{2+\sqrt{5}-1}\cdot(2+\sqrt{5})^s < 2\cdot(2+\sqrt{5})^s < 2 \cdot k^{0.3}$, as desired, since $s = 0.2\log k$ and $\log(2 + \sqrt{5}) < 1.5$ and $0.2 \cdot 1.5 = 0.3$. $\square$

We now turn to the main thrust of Lemma 17, showing that the scheme is $[O(\sqrt{\delta}), n]$-good, where $n = \delta k \log k$, and $\delta \geq \frac{1}{\log k}$; the following lemma, quantifying the "skinnyness" of the Chebyshev bumps is the cornerstone of this argument.

**Lemma 20.** $|g_2(y)| \leq \frac{\pi^7}{y^4 s^4}$ for $y \in [-\pi, \pi] \setminus (-3\pi/s, 3\pi/s)$, and $|g_2(y)| \leq 1/2$ everywhere.

*Proof.* Since $g_1(y) = \sum_{j=-s}^{s-1} \cos jy = \sin(sy) \cot(y/2)$, and since $\sin(\alpha + \pi) = -\sin(\alpha)$, we have the following:

$$
\begin{aligned}
g_2(y) &= \frac{1}{16s}\left(g_1(y - \frac{3\pi}{2s}) + 3g_1(y - \frac{\pi}{2s}) + 3g_1(y + \frac{\pi}{2s}) + g_1(y + \frac{3\pi}{2s})\right) \\
&= \frac{1}{16s}\left(\sin(ys + \pi/2)\left(\cot(\frac{y}{2} - \frac{3\pi}{4s}) - 3\cot(\frac{y}{2} - \frac{\pi}{4s})\right.\right. \\
&\qquad\qquad\qquad\qquad\left.\left. + 3\cot(\frac{y}{2} + \frac{\pi}{4s}) - \cot(\frac{y}{2} + \frac{3\pi}{4s})\right)\right).
\end{aligned}
$$

Note that $\left(\cot(\frac{y}{2} - \frac{3\pi}{4s}) - 3\cot(\frac{y}{2} - \frac{\pi}{4s}) + 3\cot(\frac{y}{2} + \frac{\pi}{4s}) - \cot(\frac{y}{2} + \frac{3\pi}{4s})\right)$ is a discrete approximation to $(\pi/2s)^3$ times the third derivative of the cotangent function evaluated at $y/2$. Thus it is bounded in magnitude by $(\pi/2s)^3$ times the maximum magnitude of $\frac{d^3}{dx^3}\cot(x)$ in the range $x \in [\frac{y}{2} - \frac{3\pi}{4s}, \frac{y}{2} + \frac{3\pi}{4s}]$. Since the magnitude of this third derivative is decreasing for $x \in (0, \pi)$, we can simply evaluate the magnitude of this derivative at $\frac{y}{2} - \frac{3\pi}{4s}$. We thus have $\frac{d^3}{dx^3}\cot(x) = \frac{-2(2 + \cos(2x))}{\sin^4(x)}$, whose magnitude is at most $\frac{6}{(2x/\pi)^4}$ for $x \in (0, \pi)$. For $y \in [3\pi/s, \pi]$, we trivially have that $\frac{y}{2} - \frac{3\pi}{4s} \geq \frac{y}{4}$, and thus we have the following bound:

$$
|\cot(\frac{y}{2} - \frac{3\pi}{4s}) - 3\cot(\frac{y}{2} - \frac{\pi}{4s}) + 3\cot(\frac{y}{2} + \frac{\pi}{4s}) - \cot(\frac{y}{2} + \frac{3\pi}{4s})| \leq \left(\frac{\pi}{2s}\right)^3 \frac{6}{(y/2\pi)^4} \leq \frac{12\pi^7}{y^4 s^3}.
$$

Since $g_2(y)$ is a symmetric function, the same bound holds for $y \in [-\pi, -3\pi/s]$. Thus $|g_2(y)| \leq \frac{12\pi^7}{16s \cdot y^4 s^3} < \frac{\pi^7}{y^4 s^4}$ for $y \in [-\pi, \pi] \setminus (-3\pi/s, 3\pi/s)$. To conclude, note that $g_2(y)$ attains a global maximum at $y = 0$, with $g_2(0) = \frac{1}{16s}\left(6\cot(\pi/4s) - 2\cot(3\pi/4s)\right) \leq \frac{1}{16s}\frac{24s}{\pi} < 1/2$. $\qquad\square$

**Lemma 21.** *The Chebyshev earthmoving scheme of Definition 16 is $[O(\sqrt{\delta}), n]$-good, where $n = \delta k \log k$, and $\delta \geq \frac{1}{\log k}$.*

*Proof.* We split this proof into two parts: first we will consider the cost of the portion of the scheme associated with all but the first $s + 1$ bumps, and then we consider the cost of the skinny bumps $f_i$ with $i \in \{0, \ldots, s\}$.

For the first part, we consider the cost of bumps $f_i$ for $i \geq s + 1$; that is the relative earthmover cost of moving $poi(xk, i)$ mass from $x$ to $\frac{i}{k}$, summed over $i \geq s$. By definition of relative earthmover distance, the cost of moving mass from $x$ to $\frac{i}{k}$ is $|\log \frac{xk}{i}|$, which, since $\log y \leq y - 1$, we bound by $\frac{xk}{i} - 1$ when $i < xk$ and $\frac{i}{xk} - 1$ otherwise. We thus split the sum into two parts.

For $i \geq \lceil xk \rceil$ we have $poi(xk, i)(\frac{i}{xk} - 1) = poi(xk, i-1) - poi(xk, i)$. This expression telescopes when summed over $i \geq \max\{s, \lceil xk \rceil\}$ to yield $poi(xk, \max\{s, \lceil xk \rceil\} - 1) = O(\frac{1}{\sqrt{s}})$.

For $i \leq \lceil xk \rceil - 1$ we have, since $i \geq s$, that $poi(xk, i)(\frac{xk}{i} - 1) \leq poi(xk, i)((1 + \frac{1}{s})\frac{xk}{i+1} - 1) = (1 + \frac{1}{s})poi(xk, i+1) - poi(xk, i)$. The $\frac{1}{s}$ term sums to at most $\frac{1}{s}$, and the rest telescopes to $poi(xk, \lceil xk \rceil) - poi(xk, s) = O(\frac{1}{\sqrt{s}})$. Thus in total, $f_i$ for $i \geq s + 1$ contributes $O(\frac{1}{\sqrt{s}})$ to the relative earthmover cost, per unit of weight moved.

We now turn to the skinny bumps $f_i(x)$ for $i \leq s$. The simplest case is when $x$ is outside the region that corresponds to the cosine of a real number — that is, when $xk \geq 4s$. It is straightforward to show that $f_i(x)$ is very small in this region. We note the general expression for Chebyshev polynomials: $T_j(x) = \frac{1}{2}\left[(x - \sqrt{x^2 - 1})^j + (x + \sqrt{x^2 - 1})^j\right]$, whose magnitude we bound by

$|2x|^j$. Further, since $2x \le \frac{2}{e}e^x$, we bound this by $(\frac{2}{e})^j e^{|x|j}$, which we apply when $|x| > 1$. Recall the definition $f_i(x) = t_i(1 - \frac{xk}{2s})\sum_{j=0}^{s-1} poi(xk, j)$, where $t_i$ is the polynomial defined so that $t_i(\cos(y)) = g_3^i(y)$, that is, $t_i$ is a linear combination of Chebyshev polynomials of degree at most $s$ and with coefficients summing in magnitude to at most 2, as was shown in the proof of Lemma 19. Since $xk > s$, we may bound $\sum_{j=0}^{s-1} poi(xk, j) \le s \cdot poi(xk, s)$. Further, since $z \le e^{z-1}$ for all $z$, letting $z = \frac{x}{4s}$ yields $x \le 4s \cdot e^{\frac{x}{4s}-1}$, from which we may bound $poi(xk, s) = \frac{(xk)^s e^{-xk}}{s!} \le \frac{e^{-xk}}{s!}(4s \cdot e^{\frac{xk}{4s}-1})^s = \frac{4^s s^s}{e^s \cdot e^{3xk/4} s!} \le 4^s e^{-3xk/4}$. We combine this with the above bound on the magnitude of Chebyshev polynomials, $T_j(z) \le (\frac{2}{e})^j e^{|z|j} \le (\frac{2}{e})^s e^{|z|s}$, where $z = (1 - \frac{xk}{2s})$ yields $T_j(z) \le (\frac{2}{e^2})^s e^{\frac{xk}{2}}$. Thus $f_i(x) \le poly(s)4^s e^{-3xk/4}(\frac{2}{e^2})^s e^{\frac{xk}{2}} = poly(s)(\frac{8}{e^2})^s e^{-\frac{xk}{4}}$. Since $\frac{xk}{4} \ge s$ in this case, $f_i$ is exponentially small in both $x$ and $s$; the total cost of this earthmoving scheme, per unit of mass above $\frac{4s}{k}$ is obtained by multiplying this by the logarithmic relative distance the mass has to move, and summing over the $s+1$ values of $i \le s$, and thus remains exponentially small, and is thus trivially bounded by $O(\frac{1}{\sqrt{s}})$.

To bound the cost in the remaining case, when $xk \le 4s$ and $i \le s$, we work with the trigonometric functions $g_3^i$, instead of $t_i$ directly. For $y \in (0, \pi]$, we seek to bound the per-unit-mass relative earthmover cost of, for each $i \ge 0$, moving $g_3^i(y)$ mass from $\frac{2s}{k}(1 - \cos(y))$ to $c_i$. (Recall from Definition 16 that $c_i = \frac{2s}{k}\left(1 - \cos(\frac{i\pi}{s})\right)$ for $i \in \{1, \dots, s\}$, and $c_0 = c_1$.) For $i \ge 1$, this contribution is at most

$$\sum_{i=1}^{s} |g_3^i(y)(\log(1 - \cos(y)) - \log(1 - \cos(\frac{i\pi}{s})))|.$$

We analyze this expression by first showing that for any $x, x' \in (0, \pi]$,

$$|\log(1 - \cos(x)) - \log(1 - \cos(x'))| \le 2|\log x - \log x'|.$$

Indeed, this holds because the derivative of $\log(1 - cos(x))$ is positive, and strictly less than the derivative of $2 \log x$; this can be seen by noting that the respective derivatives are $\frac{\sin(y)}{1-\cos(y)}$ and $\frac{2}{y}$, and we claim that the second expression is always greater. To compare the two expressions, cross-multiply and take the difference, to yield $y \sin y - 2 + 2\cos y$, which we show is always at most 0 by noting that it is 0 when $y = 0$ and has derivative $y \cos y - \sin y$, which is negative since $y < \tan y$. Thus we have that $|\log(1 - \cos(y)) - \log(1 - \cos(\frac{i\pi}{s}))| \le 2|\log y - \log \frac{i\pi}{s}|$; we use this bound in all but the last step of the analysis. Additionally, we ignore the $\sum_{j=0}^{s-1} poi(xk, j)$ term as it is always at most 1.

**Case 1:** $y \ge \frac{\pi}{s}$.

We will show that

$$|g_3^0(y)(\log y - \log \frac{\pi}{s})| + \sum_{i=1}^{s} |g_3^i(y)(\log y - \log \frac{i\pi}{s})| = O(\frac{1}{sy}),$$

where the first term is the contribution from $f_0, c_0$. For $i$ such that $y \in (\frac{(i-3)\pi}{s}, \frac{(i+3)\pi}{s})$, by the second bounds on $|g_2|$ in the statement of Lemma 20, $g_3^i(y) < 1$, and for each of the at most 6 such $i$, $|(\log y - \log \frac{\max\{1,i\}\pi}{s})| < \frac{1}{sy}$, to yield a contribution of $O(\frac{1}{sy})$. For the contribution from $i$ such that $y \le \frac{(i-3)\pi}{s}$ or $y \ge \frac{(i-3)\pi}{s}$, the first bound of Lemma 20 yields $|g_3^i(y)| = O(\frac{1}{(ys-i\pi)^4})$. Roughly, the bound will follow from noting that this sum of inverse fourth powers is dominated by the first few terms. Formally, we split up our sum over $i \in [s] \setminus [\frac{ys}{\pi} - 3, \frac{ys}{\pi} + 3]$ into two parts according to whether $i > ys/\pi$:

$$\begin{aligned}
\sum_{i \ge \frac{ys}{\pi}+3}^{s} \frac{1}{(ys - i\pi)^4}|(\log y - \log \frac{i\pi}{s})| &\le \sum_{i \ge \frac{ys}{\pi}+3}^{\infty} \frac{\pi^4}{(\frac{ys}{\pi} - i)^4}(\log i - \log \frac{ys}{\pi}) \\
&\le \pi^4 \int_{w=\frac{ys}{\pi}+2}^{\infty} \frac{1}{(\frac{ys}{\pi} - w)^4}(\log w - \log \frac{ys}{\pi}). \quad (2)
\end{aligned}$$

Since the antiderivative of $\frac{1}{(\alpha-w)^4}(\log w - \log \alpha)$ with respect to $w$ is

$$\frac{-2w(w^2 - 3w\alpha + 3\alpha^2)\log w + 2(w-\alpha)^3 \log(w-\alpha) + \alpha(2w^2 - 5w\alpha + 3\alpha^2 + 2\alpha^2 \log \alpha)}{6(w-\alpha)^3 \alpha^3},$$

the quantity in Equation 2 is equal to the above expression evaluated with $\alpha = \frac{ys}{\pi}$, and $w = \alpha + 2$, to yield

$$O(\frac{1}{ys}) - \log \frac{ys}{\pi} + \log(2 + \frac{ys}{\pi}) = O(\frac{1}{ys}).$$

A nearly identical argument applies to the portion of the sum for $i \leq \frac{ys}{\pi} + 3$, yielding the same asymptotic bound of $O(\frac{1}{ys})$.

**Case 2:** $\frac{ys}{\pi} < 1$.

The per-unit mass contribution from the 0th bump is trivially at most $|g_3^0(y)(\log \frac{ys}{\pi} - \log 1)| \leq \log \frac{ys}{\pi}$. The remaining relative earthmover cost is $\sum_{i=1}^{s} |g_3^i(y)(\log \frac{ys}{\pi} - \log i)|$. To bound this sum, we note that $\log i \geq 0$, and $\log \frac{ys}{\pi} \leq 0$ in this region, and thus split the above sum into the corresponding two parts, and bound them separately. By Lemma 20, we have:

$$\sum_{i=1}^{s} g_3^i(y)\log i \leq O\left(1 + \sum_{i=3}^{\infty} \frac{\log i}{\pi^4(i-1)^4}\right) = O(1).$$

$$\sum_{i=1}^{s} g_3^i(y)\log \frac{ys}{\pi} \leq O\left(\log ys\right) \leq O(\frac{1}{ys}),$$

since for $ys \leq \pi$, we have $|\log ys| < 4/ys$.

Having concluded the case analysis, recall that we have been using the change of variables $x = \frac{2s}{k}(1 - \cos(y))$. Since $1 - \cos(y) = O(y^2)$, we have $xk = O(sy^2)$. Thus the case analysis yielded a bound of $O(\frac{1}{ys})$, which we may thus express as $O(\frac{1}{\sqrt{sxk}})$.

For a distribution with histogram $h$, the cost of moving earth on this region, for bumps $f_i$ where $i \leq s$ is thus

$$O(\sum_{x:h(x)\neq 0} h(x) \cdot x \cdot \frac{1}{\sqrt{sxk}}) = O(\frac{1}{\sqrt{sk}} \sum_{x:h(x)\neq 0} h(x)\sqrt{x}).$$

Since $\sum_x x \cdot h(x,y) = 1$, and $\sum_x h(x) \leq n$, by the Cauchy–Schwarz inequality,

$$\sum_x \sqrt{x}h(x) = \sum_x \sqrt{x \cdot h(x)}\sqrt{h(x)} \leq \sqrt{n},$$

and hence since $n = \delta k \log k$, the contribution to the cost of these bumps is bounded by $O(\sqrt{\frac{n}{sk}}) = O(\sqrt{\delta})$. As we have already bounded the relative earthmover cost for bumps $f_i$ for $i > s$ at least this tightly, this concludes the proof. $\square$

We are now equipped to prove Theorem 2.

*Proof of Theorem 2.* Let $g$ be the generalized histogram returned by Algorithm 2, and let $h$ be the generalized histogram constructed in Lemma 12—assuming the sample from the true distribution $p$ is "faithful", which occurs with probability $1 - e^{-n^{\Omega(1)}}$ by Lemma 11. Lemma 12 asserts that $R(p,h) = O(\frac{1}{k^{\Omega(1)}})$. Let $h', g'$ be the generalized histograms that result from applying the Chebyshev earthmoving scheme of Definition 16 to $h$ and $g$, respectively. By Lemma 17, $R(h,h') = O(\sqrt{1/c})$, and $R(g,g') = O(\sqrt{1/c})$. Our goal is to bound $R(p,g)$, which we do via the triangle inequality as

$$R(p,g) \leq R(p,h) + R(h,h') + R(h',g') + R(g',g) = O(\sqrt{1/c}) + R(g',h').$$

All that remains is to prove the bound $R(g',h') = O(\frac{1}{k^{\Omega(1)}})$.

Our strategy to bound this relative earthmover distance is to construct an earthmoving scheme that equates $g'$ and $h'$ whose cost can be related to the terms of the first constraint of the linear program. By definition, $g', h'$ are generalized histograms supported at the bump centers $c_i$. Our earthmoving scheme is defined as follows: for each $i \notin [k^{\mathcal{B}}, k^{\mathcal{B}} + 2k^{\mathcal{C}}]$, if $h'(c_i) > g'(c_i)$, then we move $c_i(h'(c_i) - g'(c_i))$ units of probability mass in $h'$ from location $c_i$ to location $\frac{k^{\mathcal{B}}}{k}$; analogously, if $h'(c_i) < g'(c_i)$, then we move $c_i(g'(c_i) - h'(c_i))$ units of probability mass in $g'$ from location $c_i$ to location $\frac{k^{\mathcal{B}}}{k}$. After performing this operation, the remaining discrepancy in the resulting histograms will be confined to probability range $[\frac{k^{\mathcal{B}}}{k}, \frac{k^{\mathcal{B}}+2k^{\mathcal{C}}}{k}]$, and hence can be equated at an additional cost of at most

$$\log \frac{k^{\mathcal{B}} + 2k^{\mathcal{C}}}{k^{\mathcal{B}}} = O(k^{\mathcal{C}-\mathcal{B}}) = O(\frac{1}{k^{\Omega(1)}}).$$

We now analyze the relative earthmover cost of equalizing $h'(c_i)$ and $g'(c_i)$ for all $i \notin [k^{\mathcal{B}}, k^{\mathcal{B}}+2k^{\mathcal{C}}]$ by moving the discrepancy to location $\frac{k^{\mathcal{B}}}{k}$. Since all but the first $s+1$ bumps are simply the standard Poisson bumps $f_i(x) = poi(xk, i-1)$, for $i > s$ we have

$$|h'(c_i) - g'(c_i)| = \left| \sum_{x:h(x)+g(x)\neq 0} (h(x) - g(x))x \cdot poi(kx, i-1) \right|$$

$$= \left| \sum_{x:h(x)+g(x)\neq 0} (h(x) - g(x))poi(kx, i)\frac{i}{k} \right|.$$

Recall by construction that $h(x) = g(x)$ for all $x > \frac{k^{\mathcal{B}}+k^{\mathcal{C}}}{k}$. Thus by tail bounds for Poissons, the total relative earthmover cost of equalizing $h'$ and $g'$ for all bump centers $c_i$ with $i > k^{\mathcal{B}} + 2k^{\mathcal{C}}$ is trivially bounded by $o(\frac{\log k}{k})$.

Next, we consider the contribution of the discrepancies in the Poisson bumps with centers $c_i$ for $i \in [s+1, k^{\mathcal{B}}]$. Since $\sum_{i \leq k^{\mathcal{B}}} poi(kx, i) = o(1/k^2)$ for $x \geq \frac{k^{\mathcal{B}}+k^{\mathcal{C}}}{k}$, the discrepancy in the first $k^{\mathcal{B}}$ expected fingerprints of $g, h$ is specified, up to negligible error, by the terms in the first constraint of the linear program:

$$\sum_{i<k^{\mathcal{B}}} \left| \sum_{x:h(x)+g(x)\neq 0} (h(x) - g(x))poi(kx, i)\frac{i}{k} \right|$$

$$\leq \sum_{i<k^{\mathcal{B}}} \frac{i}{k} \cdot \frac{\sqrt{k+1}}{\sqrt{\mathcal{F}_i+1}} \left( \left| \mathcal{F}_i - \sum_{x:g(x)\neq 0} g(x)poi(kx, i) \right| + \left| \mathcal{F}_i - \sum_{x:h(x)\neq 0} h(x)poi(kx, i) \right| \right)$$

$$\leq O(k^{3\mathcal{B}-1/2}) = O(\frac{1}{k^{\Omega(1)}})$$

Finally, we consider the contribution of the discrepancies in the first $s+1 = O(\log k)$ bump centers, corresponding to the skinny Chebyshev bumps. Note that for such centers, $c_i$, the corresponding bump functions $f_i(x)$ are expressible by definition as $f_i(x) = \sum_{j\geq 0} a_{ij}poi(xk, j)$, for some coefficients $a_{ij}$, where $\sum_{j\geq 0} a_{ij} \leq \beta$. Thus we have the following, where $\sum_x$ is shorthand for

$\sum_{x:h(x)+g(x)\neq 0}$:

$$|h'(c_i) - g'(c_i)| = \left|\sum_x (h(x) - g(x))x f_i(x)\right|$$

$$= \left|\sum_x (h(x) - g(x))x \sum_{j\geq 0} a_{ij} poi(xk, j)\right|$$

$$= \left|\sum_{j\geq 0} a_{ij} \sum_x (h(x) - g(x))x poi(xk, j)\right|$$

$$= \left|\sum_{j\geq 1} a_{i,j-1} \frac{j}{k} \sum_x (h(x) - g(x)) poi(xk, j)\right|.$$

Since $a_{ij} = 0$ for $j > \log k$, and since each Chebyshev bump is a linear combination of only the first $2s < \log k$ Poisson functions, the total cost of equalizing $h'$ and $g'$ at each of these Chebyshev bump centers is bounded as

$$\beta \left|\sum_{i=1}^{\log k} \frac{i}{k} \sum_x (h(x) - g(x)) poi(xk, j)\right| |\log c_0| \log k$$

where the $|\log c_0|$ term, for $c_0$ being the first bump center, is a crude upper bound on the per-unit mass relative earthmover cost of moving the mass to probability $\frac{k^{\mathcal{B}}}{k}$, and the final factor of $\log k$ is because there are at most $s < \log k$ centers corresponding to "skinny" bumps. We bound this via the triangle inequality and an appeal to the first constraint of the linear program, as above, yielding a bound of $O(\beta k^{2\mathcal{B}} \frac{\log^3 k}{\sqrt{k}})$. Since $\beta = O(k^{0.3})$ from Lemma 17, this contribution is thus also $O(\frac{1}{k^{\Omega(1)}})$.

We have thus bounded all the parts of $R(g', h')$ by $O(\frac{1}{k^{\Omega(1)}})$, completing the proof.

$\square$

We note that what we actually proved applies rather more generally than to just Linear Program 3. As long as the second and third constraints are satisfied, then if the left hand side of the first constraint, and the objective function are *somewhat* small, similar results hold.

**Proposition 22.** *For any $c > 0$, for sufficiently large $n$, given the fingerprint $\mathcal{F}$ from a "faithful" sample of size $k = c\frac{n}{\log n}$ from a distribution $p \in \mathcal{D}^n$, consider any vector $v_x$ indexed by elements $x \in X := \{\frac{1}{k^2}, \frac{2}{k^2}, \frac{3}{k^2}, \ldots, \frac{k^{\mathcal{B}} + k^{\mathcal{C}}}{k}\}$ such that*

- $\sum_{x\in X} x \cdot v_x + \sum_{i=k^{\mathcal{B}}+2k^{\mathcal{C}}}^{k} \frac{i}{k}\mathcal{F}_i = 1$

- $\forall x \in X, v_x \geq 0$

*Let $A := \sum_{x\in X} v_x$, and let $B := \sum_{i=1}^{k^{\mathcal{B}}} \frac{1}{\sqrt{\mathcal{F}_i+1}} \left|\mathcal{F}_i - \sum_{x\in X} poi(kx, i)v_x\right|$.*

*Appending the high-frequency portion of $\mathcal{F}$ to $v$ as in Algorithm 2, returns a generalized histogram $g_{LP}$ such that*

$$R(p, g_{LP}) \leq O\left(\frac{1}{\sqrt{c}} + \sqrt{\frac{A}{k\log k}} + \frac{B\log^3 k}{k^{0.2}}\right).$$

This implies, for example, that the results of Theorem 2 hold even when the right hand side of the first constraint is increased by any constant factor, and, instead of optimizing the objective function, any point with objective less than a constant multiple of $n$ is chosen. (Of course, in practice one usually does not know $n$—the support size of the unknown distribution—so minimizing the objective function is a natural way to guarantee this criterion.)

# D   Matlab code

Below is our Matlab implementation of Algorithm 1. Our implementation uses the *linprog* command for solving the linear programs, which requires Matlab's Optimization toolkit. This code is also available from our websites.

```matlab
1   function [histx,x] = unseen(f)
2   % Input: fingerprint f, where f(i) represents number of elements that
3   % appear i times in a sample.  Thus sum_i i*f(i) = sample size.
4   % File makeFinger.m transforms a sample into the associated ...
        fingerprint.
5   %
6   % Output: approximation of 'histogram' of true distribution.  ...
        Specifically,
7   % histx(i) represents the number of domain elements that occur with
8   % probability x(i).   Thus sum_i x(i)*histx(i) = 1, as ...
        distributions have
9   % total probability mass 1.
10  %
11  % An approximation of the entropy of the true distribution can be ...
        computed
12  % as:    Entropy = (-1)*sum(histx.*x.*log(x))
13
14  f=f(:)';
15  k=f*(1:size(f,2))';  %total sample size
16
17
18  %%%%%% algorithm parameters %%%%%%%%%%%
19  gridFactor = 1.1;      % the grid of probabilities will be ...
        geometric, with this ratio.
20  % setting this smaller may slightly increase accuracy, at the cost ...
        of speed
21  alpha = .5; %the allowable discrepancy between the returned ...
        solution and the "best" (overfit).
22  % 0.5 worked well in all examples we tried, though the results ...
        were nearly indistinguishable
23  % for any alpha between 0.25 and 1.  Decreasing alpha increases ...
        the chances of overfitting.
24  xLPmin = 1/(k*max(10,k));   % minimum allowable probability.
25  % a more aggressive bound like 1/k^1.5 would make the LP slightly ...
        faster,
26  % though at the cost of accuracy
27  maxLPIters = 1000;     % the 'MaxIter' parameter for Matlab's ...
        'linprog' LP solver.
28  %%%%%%%%%%%%%%%%%%%%%%%%%%%%%%%%%%%%%%%%
29
30
31  % Split the fingerprint into the 'dense' portion for which we
32  % solve an LP to yield the corresponding histogram, and 'sparse'
33  % portion for which we simply use the empirical histogram
34  x=0;
35  histx = 0;
36  fLP = zeros(1,max(size(f)));
37  for i=1:max(size(f))
38      if f(i)>0
39          wind = ...
                [max(1,i-ceil(sqrt(i))),min(i+ceil(sqrt(i)),max(size(f)))];
40          if sum(f(wind(1):wind(2)))<2*sqrt(i)
41              x=[x, i/k];
42              histx=[histx,f(i)];
43              fLP(i)=0;
44          else
45              fLP(i)=f(i);
```

```matlab
46              end
47          end
48      end
49
50      % If no LP portion, return the empirical histogram
51      fmax = max(find(fLP>0));
52      if min(size(fmax))==0
53          x=x(2:end);
54          histx=histx(2:end);
55          return;
56      end
57
58      % Set up the first LP
59      LPmass = 1 - x*histx';  %amount of probability mass in the LP region
60
61      fLP=[fLP(1:fmax), zeros(1,ceil(sqrt(fmax)))];
62      szLPf=max(size(fLP));
63
64      xLPmax = fmax/k;
65      xLP=xLPmin*gridFactor.^(0:ceil(log(xLPmax/xLPmin)/log(gridFactor)));
66      szLPx=max(size(xLP));
67
68      objf=zeros(szLPx+2*szLPf,1);
69      objf(szLPx+1:2:end)=1./(sqrt(fLP+1));  % discrepancy in ith ...
            fingerprint expectation
70      objf(szLPx+2:2:end)=1./(sqrt(fLP+1));  % weighted by 1/sqrt(f(i) + 1)
71
72      A = zeros(2*szLPf,szLPx+2*szLPf);
73      b=zeros(2*szLPf,1);
74      for i=1:szLPf
75          A(2*i-1,1:szLPx)=poisspdf(i,k*xLP);
76          A(2*i,1:szLPx)=(-1)*A(2*i-1,1:szLPx);
77          A(2*i-1,szLPx+2*i-1)=-1;
78          A(2*i,szLPx+2*i)=-1;
79          b(2*i-1)=fLP(i);
80          b(2*i)=-fLP(i);
81      end
82
83      Aeq = zeros(1,szLPx+2*szLPf);
84      Aeq(1:szLPx)=xLP;
85      beq = LPmass;
86
87
88      options = optimset('MaxIter', maxLPIters,'Display','off');
89      for i=1:szLPx
90          A(:,i)=A(:,i)/xLP(i);    %rescaling for better conditioning
91          Aeq(i)=Aeq(i)/xLP(i);
92      end
93      [sol, fval, exitflag, output] = linprog(objf, A, b, Aeq, beq, ...
            zeros(szLPx+2*szLPf,1), Inf*ones(szLPx+2*szLPf,1),[], options);
94      if exitflag==0
95              'maximum number of iterations reached--try increasing ...
                    maxLPIters'
96      end
97      if exitflag<0
98          'LP1 solution was not found, still solving LP2 anyway...'
99          exitflag
100     end
101
102     % Solve the 2nd LP, which minimizes support size subject to ...
            incurring at most
103     % alpha worse objective function value (of the objective function ...
            in the
104     % previous LP).
105     objf2=0*objf;
```

```
106  objf2(1:szLPx) = 1;
107  A2=[A;objf'];          % ensure at most alpha worse obj value
108  b2=[b; fval+alpha];    % than solution of previous LP
109  for i=1:szLPx
110      objf2(i)=objf2(i)/xLP(i);   %rescaling for better conditioning
111  end
112  [sol2, fval2, exitflag2, output] = linprog(objf2, A2, b2, Aeq, ...
         beq, zeros(szLPx+2*szLPf,1), Inf*ones(szLPx+2*szLPf,1),[], ...
         options);
113
114  if not(exitflag2==1)
115      'LP2 solution was not found'
116      exitflag2
117  end
118
119
120  %append LP solution to empirical portion of histogram
121  sol2(1:szLPx)=sol2(1:szLPx)./xLP';   %removing the scaling
122  x=[x,xLP];
123  histx=[histx,sol2'];
124  [x,ind]=sort(x);
125  histx=histx(ind);
126  ind = find(histx>0);
127  x=x(ind);
128  histx=histx(ind);
```

```
1   function f=makeFinger(v)
2
3   % Input:  vector of integers, v
4   % Output: vector of fingerprints, f where f(i) = |{j: ...
        |{k:v(k)=j}|=i }|
5   %          i.e. f(i) is the number of elements that occur exactly i ...
        times
6   %          in the vector v
7
8   h1 = hist(v,min(v):max(v));
9   f=hist(h1,0:max(h1));
10  f=f(2:end);
```

Example of how to invoke the `unseen` estimator:

```
1   % Generate a sample of size 10,000 from the uniform distribution ...
        of support 100,000
2   n=100000; k=10000;
3   samp = randi(n,k,1);
4
5   % Compute corresponding 'fingerprint'
6   f = makeFinger(samp);
7
8
9   % Estimate distribution from which sample was drawn
10  [h,x]=unseen(f);
11
12
13  %output entropy of the true distribution, Unif[n]
14  trueEntropy = log(n)
15
16  %output entropy of the empirical distribution of the sample
17  empiricalEntropy = ...
        -f'*(((1:max(size(f)))/k).*log(((1:max(size(f)))/k)))'
18
19  %output entropy of the recovered histogram, [h,x]
20  estimatedEntropy = -h*(x.*log(x))'
```