[Reviews · NeurIPS 2013]

Submitted by Assigned_Reviewer_3

Summary:
The paper introduces an algorithm for estimating the histogram of distributions where the exact size of the support is unknown. From the histogram, properties such as the size of the support and the entropy of the distribution can be derived. As input, the algorithm takes the fingerprint of an IID sample (i.e., how many elements occur with frequency 1, 2, 3, ...). It then solves two linear programs to find the “simplest plausible” histogram that would have generated the fingerprint, where plausibility is based on the assumption that the instance frequencies are Poisson distributed and close to their expected values. In Theorem 1, the authors show their algorithm learns the entropy of distributions over finite domains with arbitrary precision. Experiments in Sec. 3 show clear improvements over state-of-the-art algorithms, and the application to estimating the number of different words in Shakespeare's Hamlet from a text subsample.

Significance/Originality:
The paper builds on work by Valiant and Valiant, so the idea is not entirely new. However, the original algorithm is modified to a two-stage linear program (aiming to find the simplest plausible histogram instead of the most plausible one), and experimental results on simulated and real data are provided. In the supplementary material, the authors show improvements of their two-stage program over the original algorithm by Valiant and Valiant. A few more details in the discussion of these results would be helpful to assess the significance, e.g., are the “quirks” in the performance of the original estimator due to fitting too large histograms?

Clarity/Quality:
Overall, the presentation of the methodology and results is very clear; the authors provide many examples which help to develop intuition.
Some of the settings in Algorithm 1 appear a bit ad-hoc:
- the criterion for distinguishing between “easy” and “hard” fingerprints (line 239)
- the choice of 1/sqrt(1+Fi) to penalize the discrepancy between empirical/fitted fingerprints
The paper provides some motivation for the choice of the penalizer but, still, there would be many other choices. Do the authors have any insights (empirical/theoretical) why their choices are preferable?
Another fact which strikes me is that Theorem 1 doesn't make any assumption on the vectors x=x1,...,xL. This would mean the estimator is consistent even if one chooses the singleton list x=1. Is this correct? It would be surprising, because in this case the output of the two linear programs would be somewhat “trivial”. In their rebuttal, the authors pointed out that there was a misunderstanding and said they would put explicit conditions on the density of the grid x1,...,xL into the final version of the paper (which were included before in the supplementary material).
Summary: Pros/Cons:
+ novel algorithm with impressive empirical performance
+ results could be of interest for a large part of the NIPS community
- some choices in the algorithm design appear a bit ad-hoc

Submitted by Assigned_Reviewer_4

The authors introduce a new estimator called UNSEEN to infer entropy and other properties of distributions from very limited data (N much smaller than D).
I don’t have much to say about this paper. I think it is good and requires essentially no improvements. The described estimator for entropy (and other properties) seems to outperform all previous estimators on a wide range of conditions, the presentation is clear and fairly accessible (even for a non-expert) and the usefulness immediately apparent. The authors provide matlab code for running the estimator.
Disclaimer: As I am no expert on the topic, I didn’t check the proof in detail.
Minor things:
- What is the run time of UNSEEN compared with others?
- The supplementary material seems a bit extensive.
- Matlab code should be downloadable in the final version.

Summary: Excellent and highly relevant paper

Submitted by Assigned_Reviewer_6

Summary: The authors develop a novel approach to estimate certain symmetric functionals of probability mass functions, such as entropy. They provide both theoretical guarantees, as well as very nice empirical results.

Quality: The paper is very well written. Admittedly, I'm unfamiliar with the perspective from which the authors seem to write. I am relatively familiar with non-parametric density estimates for discrete data, including various kernel estimators. The authors seem to compare performance to a few 'straw men', but then also a few relatively serious contenders. And, of course I realize that their is an essentially unlimited number of comparisons one could make. Nonetheless, I was surprised to not see any approaches that I was familiar with that were developed for this undersampled regime, such as kernel methods, or some recent manifold learning methods for density estimation (such as some of Al Hero's work), or any Brownian bridge type stuff for the estimating distance stuff.

Clarity: Although I was unfamiliar with several domain specific words, such as "domain elements" and "fingerprints", I found that the examples were sufficient to clarify. An illustrative figure in the appendix might help (might already be there, I didn't look).

Originality: Well, it seemed original to me, since I am wholly unfamiliar with this kind of approach! I especially enjoyed the Hamlet quote :)

Significance: It is often difficult for me to assess significance from such short papers. The problem the authors address is clearly highly significant in a number of applications. If the estimator's performance pans out in more extensive real world testing, then I imagine it will be highly significant. I imagine some other papers have made comparisons on other 'real' data (eg, not simulations), which I would like to see. I also would like to get a picture/figure of the relative computational time (and maybe memory) as a function of n, and possible intrinsic dimension of the underlying density.
Summary: The authors present a very nice new approach to estimate symmetric functionals of discrete densities, with both theoretical guarantees and impressive empirical performance. I'd love to see more extensive benchmarking on previously used data sets, comparing performance in terms of accuracy, as well as computational considerations.
Author Feedback

Author rebuttal: Thanks for your thoughtful comments/suggestions. Below we briefly respond to the main questions raised in the reviews.

Assigned_Reviewer_3:
>>ad-hoc choices of scaling parameter and easy/hard fingerprint transition:
Within the general framework of our linear programming approach to these estimation tasks, there were a few secondary decisions that had to be made along the way (e.g. how to gracefully deal with the two fingerprint regimes, what the proper scaling is, the mesh size used as the linear program support, etc), and the reviewer is correct in thinking that these details could be decided in a variety of natural ways (Appendix B of the supplementary section illustrates that the performance in practice is quite robust to varying some of these details). The specific choices we made aligned with our intuitions, and were also made with an eye towards enabling/simplifying the proof of the theoretical performance of the resulting estimators.

>>Linear program mesh size: "Theorem 1 doesn't make any assumption on the vectors x=x1,...,xL..."
This is a slight misunderstanding---Theorem 1 only applies for sufficiently dense meshes (and setting x to be a singleton will not yield anything worthwhile). We will make this more explicit in the final version---currently the algorithm just says "a fine mesh of values". We prove the theorem for x=x1,...xL being a quadratically spaced mesh starting at 1/k^2, and it follows from the proof that any finer mesh will work as well. It is also not hard to extend the proof to show that the mesh can be made significantly more coarse (up to a point), so as to ensure that the number of linear programming variables is much smaller than O(k^(0.2)), and hence the total theoretical runtime of the linear programs will be sublinear in $k$ (see the following comments on the theoretical/practical runtime of our estimators).


Assigned_Reviewer_4/6:
>>runtime of the estimator:
The runtime of the UNSEEN estimator is quite efficient, both in theory, and in practice. Although our estimators solve two linear programs, the number of variables is quite small (much smaller than the sample size). Theoretically, one can show that one can pick a coarse enough mesh of linear program variables so that the performance guarantees (of Theorem 1) still hold, while ensuring that the runtime of the UNSEEN estimator is LINEAR in the sample size (as are the runtimes of the other estimators to which we compare ours). In practice, the estimator is quite efficient---all the data for the entropy estimation performance plots (FIgure 1) were generated in a few hours on a standard desktop computer (and this involved running 500 trials for each sample size/distribution pair, with sample sizes as large as several million).


Assigned_Reviewer_6:
>> relation to density estimation approaches
The surprising thing about Theorem 1 is that our estimators are accurate even in the regime in which the sample size is still far too small to actually learn the distribution. That is, our estimators are geared towards the regime in which density estimation approaches will either fail, or must leverage assumptions on the topology of the domain. Kernel density estimation, manifold learning and Brownian bridge methods all assume that the distribution is smooth, in some sense, with respect to an underlying topology (for example Euclidean), whereas we make no such assumptions. Granted, such assumptions are natural in many real-world settings, though there are also many settings in which such assumptions are not realistic, and, it is reasonable to hope for robust estimation tools that work in general.